# Efficient and Robust Neural Combinatorial Optimization via Wasserstein-Based Coresets

**Xu Wang**[1] **& Fuyou Miao**[1,2]* **& Wenjie Liu**[1] **& Yan Xiong**[1]

[1]School of Computer Science and Technology, University of Science and Technology of China
[2]Hefei National Laboratory, University of Science and Technology of China
`worm@mail.ustc.edu.cn`
`mfy@ustc.edu.cn`
`lwj1217@mail.ustc.edu.cn`
`yxiong@ustc.edu.cn`

## Abstract

Combinatorial optimization (CO) is a fundamental tool in many fields. Many neural combinatorial optimization (NCO) methods have been proposed to solve CO problems. However, existing NCO methods typically require significant computational and storage resources, and face challenges in maintaining robustness to distribution shifts between training and test data. To address these issues, we model CO instances into probability measures, and introduce Wasserstein-based metrics to quantify the difference between CO instances. We then leverage a popular data compression technique, *coreset*, to construct a small-size proxy for the original large dataset. However, the time complexity of constructing a coreset is linearly dependent on the size of the dataset. Consequently, it becomes challenging when datasets are particularly large. Further, we accelerate the coreset construction by adapting it to the merge-and-reduce framework, enabling parallel computing. Additionally, we prove that our coreset is a good representation in theory. Subsequently, to speed up the training process for existing NCO methods, we propose an efficient training framework based on the coreset technique. We train the model on a small-size coreset rather than on the full dataset, and thus save substantial computational and storage resources. Inspired by hierarchical Gonzalez's algorithm, our coreset method is designed to capture the diversity of the dataset, which consequently improves robustness to distribution shifts. Finally, experimental results demonstrate that our training framework not only enhances robustness to distribution shifts but also achieves better performance with reduced resource requirements.

## 1 Introduction

Combinatorial optimization (CO) is a fundamental tool in many fields such as transportation (Contardo et al., 2012; Veres & Moussa, 2019), logistics (Laterre et al., 2018) and manufacturing (Froger et al., 2016; Dolgui et al., 2019; Liu et al., 2017). Numerous traditional exact (David Applegate, 2006; Optimization, 2020) or heuristic solvers (Croes, 1958; Helsgaun, 2017; Lamm et al., 2016) have been designed by experts to solve these problems. However, the real-world CO problems are widespread and diverse, and may even undergo rapid changes over time. Moreover, even for a fixed CO problem, human experts may be hindered by limited domain knowledge and computational difficulty (many of these CO problems are NP-hard). As a result, in many situations, it can be impractical to rely solely on hand-crafted methods developed by experts.

To address these challenges, numerous *Neural Combinatorial Optimization* (NCO) methods have been proposed, such as constructive heuristics methods (Khalil et al., 2017; Kool et al., 2018; Kwon et al., 2020; Hottung et al., 2020; Kim et al., 2022; Joshi et al., 2019; Fu et al., 2021; Geisler et al., 2021; Qiu et al., 2022; Sun & Yang, 2023; Luo et al., 2023; Vinyals et al., 2015; Bello et al., 2016; Nazari et al., 2018; Deudon et al., 2018; Xin et al., 2020; 2021; Kwon et al., 2021; Kim

---

*Corresponding author.

et al., 2021; Cheng et al., 2023; Drakulic et al., 2023) and improvement heuristics methods (Li et al., 2018; d O Costa et al., 2020; Wu et al., 2021; Chen & Tian, 2019; Li et al., 2023; Chen & Tian, 2019; Hottung et al., 2020; Joshi & Anand, 2022; Joshi et al., 2019). These methods learn heuristic solution strategies in a data-driven manner, thus dispensing with laborious manual design and expert knowledge; moreover, compared with traditional CO solvers, NCO methods can benefit from accelerated inference speeds by utilizing modern GPU devices.

Despite their advantages, these methods often require large training datasets, which demands substantial storage space and computational resources. Additionally, when training data and test data come from different distributions, enhancing the robustness to distribution shift (Liang et al., 2023; Sun et al., 2020) is also a challenge for existing NCO models. Therefore, training a competitive model with limited resources while ensuring its robustness to distribution shifts is an important and worthy problem to address.

To address these issues, we consider constructing a good representation, *coreset* (Ros & Guillaume, 2020), for the original huge dataset. Coreset is a popular data compression technique, which can accelerate the training process by reducing dataset size while preserving the value. Roughly speaking, coreset is a small-size proxy of the original dataset $\mathcal{Q}$ with respect to an objective; the value of the objective evaluated on coreset can closely approximate the value evaluated on $\mathcal{Q}$. Therefore, we can replace $\mathcal{Q}$ by coreset in the training phase, and thus save the storage space and computational resources significantly. Furthermore, our coreset method is inspired by hierarchical Gonzalez's algorithm (Gonzalez, 1985), and thus can capture the diversity of the dataset. Consequently, benefiting from its diversity, the model based on our coreset method shows robustness to distribution shift.

The intuition behind our coreset technique can be likened to preparing for an exam. Training neural networks is similar to practicing exercises for an exam. While the number of available exercises (i.e., data) might be vast, we cannot be trained for all the exercises with limited time and energy (i.e., storage and computational resources). Fortunately, the whole exercises are redundant; to get a high score, doing all the exercises is unnecessary, and we only need to cover all categories of exercises. Based on the above intuitions, we need a small-size representation (i.e., coreset) for the whole exercises. To this end, three key steps are required: i) exploring a proper metric to quantify the difference between CO instances; ii) constructing a coreset for the original dataset; iii) designing an efficient training framework based on the coreset technique for existing NCO models.

Many CO problems, such as the Traveling Salesperson Problem (TSP) and Maximum Independent Set (MIS), can inherently induce a graph structure. By employing graph embedding techniques, we can map these graph structures into a set of points in Euclidean space. Thus, we model CO instances as probability measures (in section 3.1). Wasserstein distance is commonly used to quantify the difference between probability measures. However, solutions to CO problems such as TSP remain invariant under rigid transformations such as translation, rotation, and reflection. In other words, a CO instance can generate multiple variants through these transformations; but they are inherently the same instance. Therefore, the distance between such instances should be zero. To capture this property, we introduce the **W**asserstein **d**istance under **r**igid transformations (RWD) to measure the difference between two CO instances.

**Our contributions:**

- First, we model CO instances as probability measures, and introduce RWD to quantify the difference between two given CO instances.

- Then, based on RWD, we design a coreset algorithm to effectively compress data for training acceleration; it saves substantial computational and storage resources. However, the time required to construct the coreset increases linearly with the size of the dataset, making it computationally expensive for extremely large datasets.

- To further accelerate coreset construction, we adapt our coreset method to merge-and-reduce framework, enabling parallel computation. Moreover, we demonstrate that our coreset is a good representation theoretically.

- Next, based on our coreset method, we propose an efficient training framework for accelerating the existing NCO training process. More specifically, we replace the original dataset with our coreset to accelerate the training process; in the inference phase, test instances are

aligned along our tree (i.e., $\mathcal{T}$ from Algorithm 1 or 2) before predicting their labels using the trained model.

- The experimental results show that our training framework exhibits better performance and enhanced robustness to distribution shifts.

## 1.1 OTHER RELATED WORKS

Here, we introduce several techniques that will be involved later.

**Graph embedding technique** represents the nodes and edges of a graph in Euclidean space. The edge information is encoded within the Euclidean distances between points, reducing the need to handle complex graph structures directly, as point-to-point information suffices. Moreover, it transforms the discrete graph into continuous Euclidean coordinates, which allows many techniques in Euclidean space to be used. Here are some widely used graph embedding methods. Laplacian Eigenmaps (Belkin & Niyogi, 2001) embed graph data into a low-dimensional Euclidean space while preserving local neighborhood relationships. Multidimensional Scaling (MDS) (Borg & Groenen, 2007) focuses on preserving pairwise distances between nodes in the graph. Isomap (Tenenbaum et al., 2000) extends MDS by incorporating geodesic distances along the manifold, making it especially useful for graphs with inherent nonlinear structures.

**Hierarchical Gonzalez's algorithm** (Murtagh & Contreras, 2012) is a variant of Gonzalez's $k$-center algorithm for addressing hierarchical clustering problem. In this approach, clusters are recursively divided at different levels of granularity, yielding a tree structure for efficient querying. This algorithm prioritizes selecting new center points that are far apart from the previously chosen ones. This strategy leads to clusters well-spread across the data, effectively capturing the diversity of the dataset. This method is commonly used for summarizing large datasets. However, its time complexity exhibits a linear dependence on the size of the dataset, making it potentially time-consuming for extremely large datasets. To mitigate this issue, we integrate merge-and-reduce (Bentley & Saxe, 1980; Har-Peled & Mazumdar, 2004) technique to construct our coreset in Algorithm 2.

## 2 PRELIMINARIES

**Notations** We define $[n] := \{1, \ldots, n\}$ and denote the vector of ones by $\mathbf{1}$. The $\ell_2$-norm is denoted by $\|\cdot\|$, and $|A|$ denotes the size of set $A$. Let $\mathbb{R}_+$ be the set of non-negative real numbers. Let $\mathcal{P}(\mathbb{R}^d)$ be the probability measure space on Euclidean space $\mathbb{R}^d$. Matrices are denoted by bold capital letters, such as $\mathbf{C}$; $C_{ij}$ is its element in the $i$-th row and $j$-th column. Similarly, we denote vectors by bold lowercase letters, such as $\mathbf{a} := (a_1, \ldots, a_n)^T \in \mathbb{R}^n$; $a_i$ is its $i$-th element.

Wasserstein distance (Peyré et al., 2017) is skilled at capturing the geometric structures of CO problems, but it is sensitive to rigid transformations. To obtain the invariance property under rigid transformations, we consider the following conception: *Wasserstein distance under rigid transformation* (RWD).

**Definition 2.1** (RWD). *Let $\mu = \sum_{i=1}^n a_i \delta_{x_i}, \nu = \sum_{j=1}^n b_j \delta_{y_j} \in \mathcal{P}(\mathbb{R}^d)$, where $\mathbf{a}, \mathbf{b} \in \mathbb{R}_+^n$ are their weight vectors and $\{x_i\}_{i\in[n]}, \{y_j\}_{j\in[n]} \subset \mathbb{R}^d$ are their locations. Then, the Wasserstein distance under rigid transformation between $\mu$ and $\nu$ is*

$$\mathcal{W}(\mu, \nu) := \left( \min_{\mathbf{P} \in \Pi(\mathbf{a}, \mathbf{b}), e \in \mathrm{E}(d)} \sum_{i=1}^n \sum_{j=1}^n P_{ij} \|x_i - e(y_j)\|^2 \right)^{1/2},$$

*where $\Pi(\mathbf{a}, \mathbf{b}) := \left\{ \mathbf{P} \in \mathbb{R}_+^{n \times n} \mid \mathbf{P1} = \mathbf{a}, \mathbf{P}^T \mathbf{1} = \mathbf{b} \right\}$ is the coupling set, $\mathrm{E}(d)$ is the euclidean group on $\mathbb{R}^d$, and $e : \mathbb{R}^d \to \mathbb{R}^d$ is the rigid transformation.*

**Remark 2.2.** *i) If we fix $e$ as identity transformation, then* RWD *is degenerated as the* Wasserstein distance*; Wasserstein distance is a metric on $\mathcal{P}(\mathbb{R}^d)$. ii)* RWD *is a (semi-)metric[1] on $\mathcal{P}(\mathbb{R}^d)$; more specifically, $(\mathcal{P}(\mathbb{R}^d), \mathcal{W})$ is a metric space.*

---

[1]For simplicity, we do not distinguish between metric and semi-metric.

Next, we formally define our coreset technique. Let

$$\ell : \mathcal{Q} \times \Theta \to \mathbb{R}_+, \quad (\mu, \theta) \mapsto \ell(\mu, \theta) \tag{1}$$

be a loss function, where $\theta \in \Theta$ is the model parameter and $\mu \in \mathcal{Q}$ denotes a CO instance. For any weighted set $A \subset \mathcal{Q}$ with weight function $w_A$, we define $\ell(A, \theta) := \sum_{\mu \in A} w_A(\mu) \cdot \ell(\mu, \theta)$.

**Definition 2.3** (Coreset). *Let $0 < \epsilon < 1$ and $\ell$ be a loss function. Let $\mathcal{Q} \subset \mathcal{P}(\mathbb{R}^d)$ be a set of measures with weight function $w_{\mathcal{Q}} : \mathcal{P}(\mathbb{R}^d) \to \mathbb{R}_+$. Let $\sum_{\mu \in \mathcal{Q}} w_{\mathcal{Q}}(\mu) = 1$. Then, a weighted set $\mathcal{S}$ with weight function $w_{\mathcal{S}}$ is an $\epsilon$-coreset of $\mathcal{Q}$ if*

$$\ell(\mathcal{S}, \theta) \in (1 \pm \epsilon) \cdot \ell(\mathcal{Q}, \theta) \quad \text{for all } \theta \in \Theta. \tag{2}$$

Then, we introduce some basic properties that will be used later. *Doubling dimension* (Chan et al., 2016) can describe the growth rate of the dataset with respect to some metric $\text{dist}$. Formally, the doubling dimension of metric space $(\mathcal{Q}, \text{dist})$ is the smallest positive integer $\mathsf{ddim}$ such that every ball in $(\mathcal{Q}, \text{dist})$ can be covered by $2^{\mathsf{ddim}}$ balls of half the radius. For example, the doubling dimension of the Euclidean space $\mathbb{R}^d$ is $\Theta(d)$.

The Lipschitz constant of a function describes how fast it can change. The loss function is $L$-Lipschitz continuous with respect to $\text{dist}$ on $\mathcal{Q}$, if $|\ell(\mu_1, \theta) - \ell(\mu_2, \theta)| \leq L \cdot \text{dist}(\mu_1, \mu_2)$ holds for all $\mu_1, \mu_2 \in \mathcal{Q}, \theta \in \Theta$.

# 3 OUR METHODS

This section introduces our methods. Section 3.1 introduces RWD to quantify the difference between two CO instances. Section 3.2 constructs a small-size coreset for accelerating the training process. Section 3.3 accelerates coreset construction process by using merge-and-reduce framework; moreover, we theoretically demonstrate that our coreset is a good representation. Finally, in Section 3.4, we present our efficient framework for existing NCO methods.

## 3.1 METRICS FOR CO INSTANCES

Many CO problems can induce graph structures. We first extract the graph structure induced by the CO instance and represent it by a graph metric space, where each point in this space reflects node-specific information, and edge relationships are captured through the corresponding shortest-path metric. We then apply graph embedding techniques (in Section 1.1) to map this graph metric space into Euclidean space, aiming to preserve inter-point distances closely. In this embedding, each node in the original graph is represented as a discrete point in Euclidean space, and edge information is encoded in Euclidean distances between these points. Ultimately, we represent the graph as a discrete set of points in Euclidean space. Henceforth, we focus on the point set data in Euclidean space.

Given two CO instances, we represent the nodes of their corresponding graph structure as $X = \{x_i\}_{i \in [n]}, Y = \{y_j\}_{j \in [n]} \subseteq \mathbb{R}^d$. Then, the CO instances are modeled as two probability measures $\mu = \sum_{i=1}^n a_i \delta_{x_i}$ and $\nu = \sum_{j=1}^n b_j \delta_{y_j}$ (with $a_i = b_j = \frac{1}{n}$ to represent equal node importance). Then, we can quantify the difference between $\mu$ and $\nu$ with metric RWD; that is, $\mathcal{W}(\mu, \nu)$, where the ground distance between $x \in X$ and $y \in Y$ is $\|x - y\|$ as in Definition 2.1.

**Remark 3.1.** *i) By graph embedding technique, the nodes and edges of a graph structure are described by the locations and their ground distances in Euclidean space. ii) These CO problems are usually invariant after imposing rigid transformations on their nodes, and RWD can capture this characteristic well. In essence, the complexity of data space is reduced under RWD metric. More specifically, we regard two CO instances as the same instance if their distance is zero. iii) The complete graph in $\mathbb{R}^d$ can be represented directly by a point set, without the need for graph embedding techniques. iv) If we fixed the outer iteration number and the dimension $d$ of data space as constants, our RWD can be solved within $\widetilde{\mathcal{O}}(n^2)$ time by using the heuristic method in Algorithm 3.*

## 3.2 CORESET

Intuitively, our coreset aims to cover all the data by using relatively fewer and smaller balls, where all the balls have the same radius. This strategy can be likened to preparing for an exam, where

we need to cover as many categories of exercises as possible with limited time and energy. For simplicity, we take RWD as an example to illustrate our methods. The metric RWD can also be replaced by Wasserstein distance, or some other proper metrics on $\mathcal{P}(\mathbb{R}^d)$.

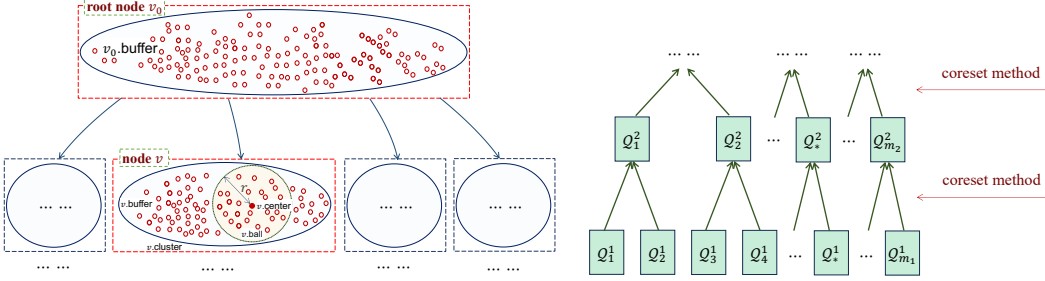

Figure 1: Grow nodes from root $v_0$ with $\mathsf{ddim} = 1$. The red dots denote probability measures in dataset $\mathcal{Q}$, and the solid red dot is the center of the ball $v.\mathsf{ball}$.

Figure 2: Accelerate the coreset construction by using merge-and-reduce framework.

**Coreset construction**   Our algorithm is inspired by the hierarchical structure in (Ding et al., 2021; Krauthgamer & Lee, 2004; Har-Peled & Mendel, 2005; Beygelzimer et al., 2006). Given a set $\mathcal{Q}$ of probability measures, we aim at clustering similar measures into small balls of radius $r$, and take the cluster centers to form our coreset. The coreset is finally reserved in the tree $\mathcal{T}$ as shown in Figure 1.

To construct such a coreset, we first initialize an empty tree $\mathcal{T}$, and set its root node as $v_0$. The root node has only one attribute $\mathsf{buffer}$, and is initialized as $v_0.\mathsf{buffer} = \mathcal{Q}$. Our (non-root) node has four attributes: $\mathsf{cluster}$, $\mathsf{center}$, $\mathsf{buffer}$ and $\mathsf{ball}$ as shown in Figure 1. The nodes grow in an up-bottom manner recursively. Given a current node $v$, if $v.\mathsf{buffer}$ is an empty set, then $v$ is a leaf node, and we stop adding children to it. If $v.\mathsf{buffer}$ is a nonempty set, we add $k = \min\{|v.\mathsf{buffer}|, 2^{2\cdot\mathsf{ddim}}\}$ children node $\{v_j'\}_{j\in[k]}$ to the current node $v$; more specifically, we run Gonzalez's algorithm $k$ rounds on $v.\mathsf{buffer}$. By this, we obtain $k$ cluster centers $v_j'.\mathsf{center}$ and their corresponding clusters $v_j'.\mathsf{cluster}$. All the $v_j'.\mathsf{cluster}$ form a partition of $v.\mathsf{buffer}$; each $v_j'.\mathsf{cluster}$ consists of points that are relatively close to its center $v_j'.\mathsf{center}$. For each set $v_j'.\mathsf{cluster}$, we partition it into two sets $v_j'.\mathsf{ball}, v_j'.\mathsf{buffer}$, where $v_j'.\mathsf{ball}$ is a RWD-based ball of radius $r$ centered at measure $v_j'.\mathsf{center}$; formally, we can formulate them as

$$v_j'.\mathsf{ball} = \left\{\mu \in v_j'.\mathsf{cluster} \mid \mathcal{W}(\mu, v_j'.\mathsf{center}) \leq r\right\} \tag{3}$$

and

$$v_j'.\mathsf{buffer} = v_j'.\mathsf{cluster} - v_j'.\mathsf{ball}. \tag{4}$$

Finally, we obtain a tree $\mathcal{T}$. The coreset $\mathcal{S}$ consists of all the center points $v.\mathsf{center}$ with weight $|v.\mathsf{ball}|$. We show the coreset construction process in a more intuitive and comprehensible manner in Figure 1. The detailed descriptions are in Algorithm 1.

**Time complexity and coreset size**   From (Ding et al., 2021), we know that the radius of the clusters will be halved after carrying out at most $2^{2\cdot\mathsf{ddim}}$ rounds of Gonzalez's algorithm. Let $R$ be the radius of dataset $\mathcal{Q}$; that is, $\mathcal{W}(\mu, \nu) \leq 2R$ for any $\mu, \nu \in \mathcal{Q}$. Thus, the height of the tree in Algorithm 1 is at most $\mathcal{O}(\log \frac{R}{r})$. Let $T(n)$ be the time for computing the distance (i.e., RWD) between two measures, where $n$ is the size of the locations of measures. Since constructing every layer takes $\mathcal{O}(2^{2\cdot\mathsf{ddim}}) \cdot |\mathcal{Q}|$ computations of RWD, the total time complexity for Algorithm 1 is $\mathcal{O}(2^{2\cdot\mathsf{ddim}}) \cdot |\mathcal{Q}| \cdot T(n) \cdot \log \frac{R}{r}$. Its time complexity increases linearly with the size of the dataset, making it computationally expensive for large datasets. The coreset is maintained in the tee $\mathcal{T}$. The tree has $\mathcal{O}((2^{2\cdot\mathsf{ddim}})^{\log \frac{R}{r}})$ nodes, thus the coreset size is $\mathcal{O}((\frac{R}{r})^{2\cdot\mathsf{ddim}})$.

**Remark 3.2.** *i) The output of Algorithm 1 contains a tree $\mathcal{T}$ and coreset $\mathcal{S}$. The $\mathcal{S}$ is a representation of the original measure set $\mathcal{Q}$, which is used for speeding up the training process for existing NCO methods; while the tree $\mathcal{T}$ is prepared for aligning CO instances at the inference phase. ii) It is often not necessary to know the exact value of the doubling dimension in advance. Typically, we*

---

**Algorithm 1** Algorithm for constructing coresets

---

**Input:** a set $\mathcal{Q} := \{\mu_i\}_{i\in[N]} \subset \mathcal{P}(\mathbb{R}^d)$ of measures, doubling dimension $\mathsf{ddim}$ of $\mathcal{Q}$, radius $r$

1: Initialize an empty tree $\mathcal{T}$, and set its root node as $v_0$;
2: Set $v_0.\mathsf{buffer} = \mathcal{Q}$;
    ▷ The root node $v_0$ only has an attribute $\mathsf{buffer}$, and it is not associated with any node.
3: Construct the nodes of $\mathcal{T}$ recursively as follows:    ▷ $v$ is the current node.
4: **if** $v.\mathsf{buffer}$ is $\emptyset$ **then**
5:    The current node $v$ is a leaf node, and we stop adding children to it;
6: **else**
7:    Set $k = \min\{|v.\mathsf{buffer}|, 2^{2\cdot\mathsf{ddim}}\}$ and add $k$ children node $\{v'_j\}_{j\in[k]}$ to the current node $v$;
8:    Run Gonzalez's algorithm $k$ rounds on $v.\mathsf{buffer}$. For each children node $v'_j$, we set its attributions $\mathsf{cluster}, \mathsf{center}, \mathsf{ball}, \mathsf{buffer}$ according to Equation (3) and Equation (4);
9: **end if**
10: Set $\mathcal{S} = \{v.\mathsf{center} \mid v$ is a node of $\mathcal{T}\}$ and set the weight as $w_{\mathcal{S}}(\mu) = |v.\mathsf{ball}|$;

**Output:** $\mathcal{T}, \mathcal{S}$

---

*begin by experimenting with relatively small values, as demonstrated in our study where we set the low doubling dimension as $ddim = 1$. In practice, even if we cannot rigorously prove that the data satisfies low doubling dimension assumption, this generally does not impact the effectiveness of our experimental results. iii) All the $v.\mathsf{ball}$ consist of a partition of $\mathcal{Q}$.*

### 3.3 ACCELERATE THE CORESET CONSTRUCTION PROCESS

In this subsection, we adapt our coreset method to merge-and-reduce framework (Bentley & Saxe, 1980; Har-Peled & Mazumdar, 2004; Wang et al., 2021); by this, we offer a technique to accelerate our coreset construction process by achieving parallel computing; moreover, it can also be used to tackle streaming data.

Algorithm 2 is a combination of our coreset method and the merge-and-reduce framework as shown in Figure 2. We first set $s = \mathcal{O}((\frac{R}{r})^{2\cdot\mathsf{ddim}})$, $H = \log_{\frac{\tau}{s}} \frac{|\mathcal{Q}|}{s}$ and $r' = \frac{r}{H}$. The height of the tree in Figure 2 is at most $H$, and it is generated in a bottom-up manner.

We perform reduce and merge procedures alternatively in each epoch. More specifically, *reduce* means data compression; that is, we run Algorithm 1 by taking $(\mathcal{Q}_i^h, \mathsf{ddim}, r')$ as input, and obtain the corresponding coreset $\mathcal{S}_i^h$; *merge* means putting together the coresets $\mathcal{S}_i^h$; that is, $\mathcal{Q}^{h+1} = \cup_{i\in[m_h]}\mathcal{S}_i^h$. After $H$ epochs, we obtain the coreset $\mathcal{S} = \mathcal{Q}^{H+1}$ and its corresponding tree $\mathcal{T} = \mathcal{T}^{H+1}$.

---

**Algorithm 2** Algorithm for accelerating the coreset construction process

---

**Input:** a set $\mathcal{Q} := \{\mu_i\}_{i\in[N]} \subset \mathcal{P}(\mathbb{R}^d)$ of measures, doubling dimension $\mathsf{ddim}$ of $\mathcal{Q}$, radius $r$

1: Set $s = \mathcal{O}((\frac{R}{r})^{2\cdot\mathsf{ddim}})$, $H = \log_{\frac{\tau}{s}} \frac{|\mathcal{Q}|}{s}$ and $r' = \frac{r}{H}$, $\mathcal{Q}^0 = \mathcal{Q}$;
2: **for** $h = 1, \ldots, H$ **do**
3:    ▷ reduce procedure
      Partition $\mathcal{Q}^h$ as $\mathcal{Q}^h = \sqcup_{i\in[m_h]}\mathcal{Q}_i^h$ with $|\mathcal{Q}_i^h| = \mathcal{O}(\tau)$ and $m_h = \lceil \frac{|\mathcal{Q}^h|}{\tau} \rceil$;
      For every $\mathcal{Q}_i^h$, we run Algorithm 1 by taking $(\mathcal{Q}_i^h, \mathsf{ddim}, r')$ as input, and output $(\mathcal{T}_i^h, \mathcal{S}_i^h)$;
4:    ▷ merge procedure
5:    $\mathcal{Q}^{h+1} = \cup_{i\in[m_h]}\mathcal{S}_i^h$;
6: **end for**
7: Set $\mathcal{S} = \mathcal{Q}^{H+1}$ and $\mathcal{T} = \mathcal{T}^{H+1}$;

**Output:** $\mathcal{T}, \mathcal{S}$

---

**Time complexity**    Given that the input size of Algorithm 1 is $\mathcal{O}(\tau)$, and its output size is $s$. Then, the tree induced by the merge-and-reduce framework has at most $H = \log_{\frac{\tau}{s}} \frac{|\mathcal{Q}|}{s}$ layers.

Each layer of Algorithm 2 performs multiple computations of Algorithm 1 in parallel. Hence, the time complexity of per layer is $\tilde{O}(2^{2 \cdot \mathsf{ddim}} \cdot \tau \cdot T(n)) \cdot \log \frac{R}{r}$. Consequently, the overall time complexity of Algorithm 2 is $\tilde{O}(2^{2 \cdot \mathsf{ddim}} \cdot \tau \cdot T(n) \cdot \log \frac{R}{r} \cdot \log_{\frac{\tau}{s}} \frac{|\mathcal{Q}|}{s})$, which is independent on the dataset size.

**Communication complexity**   Our coreset is a subset of the original dataset, allowing us to transmit only the indexes of the CO instance items rather than the data items themselves. This significantly reduces transmission costs. As a result, the additional transfer complexity introduced by our merge-and-reduce framework in Algorithm 2 is, in practice, minimal and unlikely to pose a substantial overhead.

Moreover, the coreset size of Algorithm 2 remains consistent with that in Algorithm 1, it is sufficient to retain only the tree structure of the final layer in practice.

**A good representation**   Next, we show that the coreset $\mathcal{S}$ is a good representation of the original huge dataset $\mathcal{Q}$ in the following theorem.

**Theorem 3.3.** *Assume $\ell$ is L-Lipschitz continuous on $(\mathcal{Q}, \mathsf{RWD})$ and there exists $\gamma \in \mathbb{R}_+$ such that $\ell(\mathcal{Q}, \theta) \geq \gamma$ for all $\theta \in \Theta$. Let $\mathsf{ddim}$ be the doubling dimension of $\mathcal{Q}$ with respect to $\mathsf{RWD}$, and $R$ be the radius of $\mathcal{Q}$. Then, by setting $r = \frac{\epsilon \gamma}{L}$, both Algorithm 1 and Algorithm 2 can generate an $\mathcal{O}((\frac{R}{r})^{2 \cdot \mathsf{ddim}})$ size $\epsilon$-coreset $\mathcal{S}$ for the dataset $\mathcal{Q}$; that is, for every $\theta$, it holds that $\ell(\mathcal{S}, \theta) \in (1 \pm \epsilon) \cdot \ell(\mathcal{Q}, \theta)$.*

It shows that for every parameter $\theta \in \Theta$, the value of the loss function $\ell$ evaluated on small-size coreset $\mathcal{S}$ can approximate the value on the original dataset $\mathcal{Q}$ within $\mathcal{O}(\epsilon)$ relative error. Therefore, Theorem 3.3 demonstrates that our small-size coreset $\mathcal{S}$ is a good representation for the original huge dataset $\mathcal{Q}$ with respect to the objective $\ell$.

*Proof.* Due to limited space, we only give the proof sketch here. More details are in Appendix. First, we prove that by taking $(\mathcal{Q}, \mathsf{ddim}, r)$ as input, the coreset constructed by Algorithm 2 can cover the dataset $\mathcal{Q}$ by small balls of radius $r$ by using mathematical induction. Then, we obtain that the value difference of the loss function between the data itself and its representation is small by Lipschitz continuous property. Third, by setting some parameters properly, we turn the additive error into a relative error and obtain an $\epsilon$-coreset $\mathcal{S}$. $\qquad\square$

## 3.4   AN EFFICIENT FRAMEWORK

Here, we introduce an efficient framework to train a comparative model by using limited resources for existing NCO methods. We first feed the original dataset $\mathcal{Q}$ into Algorithm 1 or Algorithm 2, and obtain the coreset $\mathcal{S}$ and tree $\mathcal{T}$. The original dataset $\mathcal{Q}$ is replaced by small-size coreset $\mathcal{S}$ in the training phase. Thus, it saves the storage and computing resources significantly in the training phase. The probability measure $\mu$ in our coreset $\mathcal{S}$ has its own weight $w_{\mathcal{S}}(\mu)$, which helps it represent the original dataset well. However, to capture the diversity better, we usually regard these data as equally important; that is, to improve the robustness to distribution shifts in experiments, we set their weights as $w_{\mathcal{S}} = \frac{1}{|\mathcal{S}|}$.

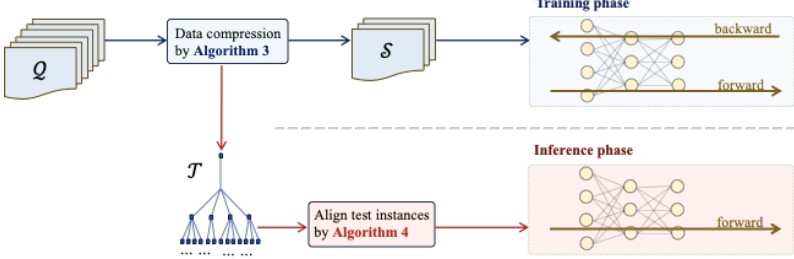

Figure 3: An efficient framework for accelerating existing NCO methods. The Algorithm 1 can be replaced by Algorithm 2, and Algorithm 4 is in Appendix.

Meanwhile, in the inference phase, we first align the test instances $\mu$ along our tree $\mathcal{T}$, which aims to find a rigid transformation such that

$$\min_{e, \nu \in \mathcal{S}} W(e(\mu), \nu), \tag{5}$$

where $W(\cdot, \cdot)$ is the Wasserstein distance and $e(\mu) := \sum_{i \in [n]} a_i \delta_{e(x_i)}$ for any $\nu = \sum_{j \in [n]} b_j \delta_{y_j}$. We offer a heuristic method (i.e., Algorithm 4 in Appendix) for the alignment as described in Equation (5). Thanks to the tree structure maintained by $\mathcal{T}$, we can finish the alignment for a test instance within $\mathcal{O}(k \cdot \log(|\mathcal{S}|) \cdot T(n))$ time. This is particularly efficient in practice since $k$ is usually small. Without this tree structure, we would potentially need to align the test instance with every training data, which could be significantly more time-consuming.

**Remark 3.4.** *By combining our coreset technique in the training phase and the alignment process in the test phase, we essentially reduce the complexity of data. Intuitively, if the model can solve an instance well, then it can solve similar instances well under metric RWD. Furthermore, our framework can be applied to other problems that inherently involve a graph structure. This demonstrates its general applicability across various domains where graph-based analysis is pertinent.*

**Remark 3.5.** *i) Our coreset only needs to be computed once, after which it can be used repeatedly to train different models and fine-tune parameters. ii) Even if the coreset computation is time-consuming, it is still valuable as it helps save storage space. iii) Our alignment process serves as an optional enhancement to improve performance rather than a mandatory step.*

Table 1: Comparison of uniform sampling and our coreset method using TSP100-2D-$\mathcal{N}(0, 1)$ as the training dataset on test data TSP100-2D from different distributions.

| Sample size | Method | Test distribution | Greedy | | Greedy+2-opt | |
|---|---|---|---|---|---|---|
| | | | Length ($\downarrow$) | Time ($\downarrow$) | Length ($\downarrow$) | Time ($\downarrow$) |
| 128000 | Org | $\mathcal{N}(0, 1)$ | 20.39 | 386 | 18.61 | 384 |
| | | $\mathcal{N}(0, 4^2)$ | 76.41 | 374 | 67.39 | 388 |
| | | $\mathcal{U}(0, 10)$ | 89.29 | 372 | 79.82 | 385 |
| 4003 | US | $\mathcal{N}(0, 1)$ | 22.34 | 378 | 18.92 | 387 |
| | | $\mathcal{N}(0, 4^2)$ | 101.95 | 379 | 69.28 | 395 |
| | | $\mathcal{U}(0, 10)$ | 119.78 | 380 | 82.59 | 395 |
| | CS | $\mathcal{N}(0, 1)$ | 22.21 | 372 | **18.87** | 379 |
| | | $\mathcal{N}(0, 4^2)$ | **80.63** | 372 | 67.92 | 379 |
| | | $\mathcal{U}(0, 10)$ | **94.73** | 373 | 80.64 | 377 |
| | CS-aligned | $\mathcal{N}(0, 1)$ | **22.18** | 359 | 18.88 | 363 |
| | | $\mathcal{N}(0, 4^2)$ | 80.66 | 362 | **67.91** | 358 |
| | | $\mathcal{U}(0, 10)$ | 94.94 | 361 | **80.53** | 360 |
| 8245 | US | $\mathcal{N}(0, 1)$ | 22.12 | 377 | 18.87 | 388 |
| | | $\mathcal{N}(0, 4^2)$ | 83.17 | 377 | 68.13 | 378 |
| | | $\mathcal{U}(0, 10)$ | 97.31 | 377 | 80.80 | 387 |
| | CS | $\mathcal{N}(0, 1)$ | **21.79** | 366 | **18.84** | 383 |
| | | $\mathcal{N}(0, 4^2)$ | 78.72 | 372 | **67.79** | 378 |
| | | $\mathcal{U}(0, 10)$ | **92.99** | 374 | **80.35** | 377 |
| | CS-aligned | $\mathcal{N}(0, 1)$ | 21.80 | 360 | 18.86 | 359 |
| | | $\mathcal{N}(0, 4^2)$ | **78.50** | 361 | 67.82 | 358 |
| | | $\mathcal{U}(0, 10)$ | 93.04 | 355 | 80.42 | 361 |
| 12951 | US | $\mathcal{N}(0, 1)$ | 21.99 | 390 | 18.87 | 377 |
| | | $\mathcal{N}(0, 4^2)$ | 80.78 | 384 | 67.94 | 379 |
| | | $\mathcal{U}(0, 10)$ | 95.01 | 369 | 80.60 | 379 |
| | CS | $\mathcal{N}(0, 1)$ | 21.57 | 372 | 18.81 | 382 |
| | | $\mathcal{N}(0, 4^2)$ | 77.80 | 369 | 67.58 | 379 |
| | | $\mathcal{U}(0, 10)$ | **92.01** | 378 | 80.23 | 375 |
| | CS-aligned | $\mathcal{N}(0, 1)$ | **21.50** | 361 | **18.79** | 358 |
| | | $\mathcal{N}(0, 4^2)$ | **77.67** | 362 | **67.57** | 357 |
| | | $\mathcal{U}(0, 10)$ | **92.01** | 358 | **80.21** | 359 |

## 4 EXPERIMENTS WITH TSP

We take TSP100 as an example to show the advantages of our coreset method. All experiments are conducted on an NVIDIA L20 GPU. Due to limited space, further experiments (including TSP training on uniformly sampled data (Kool et al., 2018), the MIS problem (Ahn et al., 2020)) and Capacitated Vehicle Routing Problem (CVRP) (Nazari et al., 2018) are presented in the Appendix [2].

Table 2: Comparison of uniform sampling and our coreset method using TSP100-2D-$\mathcal{N}(0, 1)$ as the training dataset on test data of varying sizes. We fix the sample size as 12951.

| TSP size | Method | Test distribution | Greedy | | Greedy+2-opt | |
|---|---|---|---|---|---|---|
| | | | Length ($\downarrow$) | Time ($\downarrow$) | Length ($\downarrow$) | Time ($\downarrow$) |
| TSP200 | US | $\mathcal{N}(0, 1)$ | 33.69 | 109 | 27.14 | 112 |
| | | $\mathcal{N}(0, 4^2)$ | 125.99 | 108 | 96.70 | 112 |
| | | $\mathcal{U}(0, 10)$ | 145.41 | 109 | 113.39 | 112 |
| | CS | $\mathcal{N}(0, 1)$ | **30.75** | 107 | 26.69 | 110 |
| | | $\mathcal{N}(0, 4^2)$ | **110.48** | 109 | 94.84 | 111 |
| | | $\mathcal{U}(0, 10)$ | 129.77 | 107 | **111.47** | 109 |
| | CS-aligned | $\mathcal{N}(0, 1)$ | 30.77 | 77 | **26.68** | 79 |
| | | $\mathcal{N}(0, 4^2)$ | 110.99 | 78 | **94.59** | 79 |
| | | $\mathcal{U}(0, 10)$ | **129.28** | 76 | 111.49 | 78 |
| TSP500 | US | $\mathcal{N}(0, 1)$ | 59.81 | 1012 | 43.41 | 1020 |
| | | $\mathcal{N}(0, 4^2)$ | 237.72 | 1012 | 154.28 | 1022 |
| | | $\mathcal{U}(0, 10)$ | 263.66 | 1015 | 180.75 | 1022 |
| | CS | $\mathcal{N}(0, 1)$ | **49.11** | 1012 | **42.25** | 1016 |
| | | $\mathcal{N}(0, 4^2)$ | **178.56** | 1010 | **149.50** | 1016 |
| | | $\mathcal{U}(0, 10)$ | **208.36** | 1011 | **174.93** | 1016 |
| | CS-aligned | $\mathcal{N}(0, 1)$ | 49.38 | 680 | 42.26 | 683 |
| | | $\mathcal{N}(0, 4^2)$ | 178.88 | 679 | 149.63 | 682 |
| | | $\mathcal{U}(0, 10)$ | 208.77 | 678 | 175.03 | 682 |
| TSP1000 | US | $\mathcal{N}(0, 1)$ | 94.71 | 2823 | 61.72 | 2848 |
| | | $\mathcal{N}(0, 4^2)$ | 382.77 | 4224 | 219.16 | 2847 |
| | | $\mathcal{U}(0, 10)$ | 426.61 | 4215 | 255.95 | 4254 |
| | CS | $\mathcal{N}(0, 1)$ | **69.76** | 2823 | 59.59 | 2833 |
| | | $\mathcal{N}(0, 4^2)$ | **252.92** | 4224 | **210.71** | 2832 |
| | | $\mathcal{U}(0, 10)$ | **299.80** | 4215 | **246.57** | 4234 |
| | CS-aligned | $\mathcal{N}(0, 1)$ | 69.96 | 2825 | **59.57** | 2832 |
| | | $\mathcal{N}(0, 4^2)$ | 253.63 | 2821 | 210.81 | 2830 |
| | | $\mathcal{U}(0, 10)$ | 300.03 | 2812 | 246.61 | 2827 |

**Dataset** We apply our method on TSP100-2D/3D Euclidean instances. The labels of TSP100-2D instances are obtained by using the LKH-3 heuristic solver (Helsgaun, 2017); each coordinate of the nodes in a TSP instance is generated by $x\%10$, where $x$ is randomly sampled either from a normal distribution $\mathcal{N}(0, \sigma^2)$ or uniform sampling $\mathcal{U}(0, 10)$. Our training data, TSP100-2D-$\mathcal{N}(0, 1)$ and TSP100-3D-$\mathcal{N}(0, 1)$, consists of 125,000 instances generated by the normal distribution $\mathcal{N}(0, 1)$ and 3000 instances by the uniform distribution $\mathcal{U}(0, 10)$. Indeed, the distribution of the training dataset is very close to the normal distribution $\mathcal{N}(0, 1)$. Hence, we regard the test data sampled from distribution $\mathcal{N}(0, 1)$ as having no distribution shift. The test data are sampled from a single distribution, either a normal distribution or uniform distribution. Specifically, we sample 1280 test data items for TSP100, and 128 test data items for other cases. Obviously, the uniform distribution has the highest entropy and thus the highest diversity. For these Gaussian distributions, the larger the variance, the larger the diversity.

As for the TSP100-3D dataset, we can directly extend the 2D instances to 3D instances by appending a third coordinate with a value of zero. Let $X = \{x_i\}_{i \in [n]} \subset \mathbb{R}^3$ are the nodes of an instance. We apply random rotation transformation $e$ on $X$; that is, $e(X) := \{e(x_i)\}_{i \in [n]}$.

---

[2]Code can be found at Coreset2025.

**Setting** We use the DIFUSCO (Sun & Yang, 2023) as our NCO solver. The detailed parameter settings are in Appendix. To quantify the performance of different methods, we use two criteria: the average tour length (Length) and the total runtime (Time). The term "Greedy" refers to the greedy decoding method of DIFUSCO, and "2-opt" is a post-processing used to improve solutions. The terms US,CS and CS-aligned represent uniform sampling, our coreset without alignment, and our coreset with alignment, respectively. We take the model trained on the full dataset with 128000 data items as baseline.

Table 3: Comparison of uniform sampling and our coreset method using TSP100-2D-$\mathcal{N}(0,1)$ as the training dataset on test data TSPLIB(Reinelt, 1991).

| Sample size | Method | Test distribution | Greedy | | Greedy+2-opt | |
|---|---|---|---|---|---|---|
| | | | Length ($\downarrow$) | Time ($\downarrow$) | Length ($\downarrow$) | Time ($\downarrow$) |
| 128000 | Org | $\mathcal{N}(0,1)$ | 129.35 | 108 | 112.23 | 106 |
| 4003 | US | $\mathcal{N}(0,1)$ | 190.79 | 109 | 115.87 | 108 |
| | CS | $\mathcal{N}(0,1)$ | 153.08 | 107 | **113.56** | 108 |
| | CS-aligned | $\mathcal{N}(0,1)$ | **152.70** | 103 | 113.71 | 105 |
| 8245 | US | $\mathcal{N}(0,1)$ | 166.40 | 106 | 114.47 | 108 |
| | CS | $\mathcal{N}(0,1)$ | 140.49 | 107 | 113.04 | 106 |
| | CS-aligned | $\mathcal{N}(0,1)$ | **140.18** | 104 | **112.91** | 104 |
| 12951 | US | $\mathcal{N}(0,1)$ | 162.19 | 107 | 114.31 | 107 |
| | CS | $\mathcal{N}(0,1)$ | 133.63 | 106 | **112.45** | 105 |
| | CS-aligned | $\mathcal{N}(0,1)$ | **133.14** | 103 | 112.52 | 110 |

**Results of TSP100-2D** Tables 1 to 3 present the results on training dataset TSP100-2D-$\mathcal{N}(0,1)$. The training datasets are generated by the uniform sampling and our coreset technique respectively. The results show that both our method and uniform sampling method perform better as the sample size increases. Meanwhile, as the sample size decreases, the advantage of our methods compared to uniform sampling becomes increasingly evident.

Moreover, Tables 1 and 2 demonstrate that our method is robust to distribution shift. Specifically, the training data is sampled from a normal distribution $\mathcal{N}(0,1)$, while the test data are sampled from normal distributions $\mathcal{N}(0,1)$, $\mathcal{N}(0,4^2)$ and a uniform distribution $\mathcal{U}(0,10)$. The test distributions $\mathcal{N}(0,4^2)$ and $\mathcal{U}(0,10)$ are significantly different from the training distribution $\mathcal{N}(0,1)$, which represent substantial distribution shifts. The results in Tables 1 and 2 show that our method consistently outperforms the baselines, demonstrating its robustness to distribution shifts.

Furthermore, Table 2 shows that models trained on our coreset can generalize better to larger problem sizes such as TSP200, TSP500 and TSP1000. Moreover, Table 3 confirms that our method outperforms other approaches on the TSPLIB dataset.

**Results of TSP100-3D** Tables 8 to 10 show the results on training dataset TSP100-3D-$\mathcal{N}(0,1)$. For the test data without distribution shift (i.e., $\mathcal{N}(0,1)$), our method has comparable performance; for the test data occurring distribution shift (i.e., $\mathcal{N}(0,4^2)$ and $\mathcal{U}(0,10)$), our method has better performance. Moreover, our coreset with alignment version performs better in the TSP-3D case. From Tables 1 to 3, 9 and 10, alignment can perform better for higher dimension dataset (i.e., TSP-3D). Thus, the alignment version is promising for tackling high-dimensional data. (The details are in Appendix.)

## 5 CONCLUSION AND FUTURE WORK

In this paper, we introduce an efficient training framework for NCO problems based on our coreset method. More specifically, we replace the original huge dataset with our coreset during the training phase. In the test phase, we first align the test instances with the data in our coreset, and then feed them into existing NCO models. Our framework enables the development of comparable models with limited computational and storage resources; additionally, it exhibits robustness to distribution shifts. Moreover, in future work, we will extend our method to other situations that can induce graph structures.

## 6 ACKNOWLEDGEMENTS

I would like to express my sincere gratitude to ChatGPT for assisting in refining the language and clarity of this paper. I am also deeply grateful for the support from Innovation Program for Quantum Science and Technology (2021ZD0302902), Hi-tech project(231-08-01) and Anhui Province University Natural Science Research Project (2023AH051102).

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

## A OTHER PRELIMINARIES

**Lemma A.1** (Generalized triangle inequalities(Makarychev et al., 2022)). *Given three points $a, b, c$, the following inequalities hold for any $0 < t \le 1$:*

- $\text{dist}^2(a, b) \le (1 + t) \cdot \text{dist}^2(a, c) + (1 + \frac{1}{t}) \cdot \text{dist}^2(b, c)$;

- $\left| \text{dist}^2(a, c) - \text{dist}^2(b, c) \right| \le t \cdot \text{dist}^2(a, c) + \frac{6}{t} \cdot \text{dist}^2(a, b)$.

**Definition A.2** (Wasserstein distance (Peyré et al., 2017)). *Let $\mu = \sum_{i=1}^{n} a_i \delta_{x_i}, \nu = \sum_{j=1}^{n} b_j \delta_{y_j} \in \mathcal{P}(\mathbb{R}^d)$, where $\mathbf{a}, \mathbf{b} \in \mathbb{R}_+^n$ are their weights and $\{x_i\}_{i \in [n]}, \{y_j\}_{j \in [n]} \subset \mathbb{R}^d$ are their locations. Given a cost matrix $\mathbf{C} \in \mathbb{R}_+^{n \times n}$ with $C_{ij} = \|x_i - y_j\|^2$, the Wasserstein distance between $\mu$ and $\nu$ is*

$$W(\mu, \nu) := \left( \min_{\mathbf{P} \in \Pi(\mathbf{a}, \mathbf{b})} \langle \mathbf{P}, \mathbf{C} \rangle \right)^{1/2},$$

*where $\Pi(\mathbf{a}, \mathbf{b}) := \left\{ \mathbf{P} \in \mathbb{R}_+^{n \times n} \mid \mathbf{P}\mathbf{1} = \mathbf{a}, \mathbf{P}^T\mathbf{1} = \mathbf{b} \right\}$ is the coupling set and $\mathbf{1}$ is the vector of ones.*

### A.1 OTHER RELATED WORKS

**NCO** The existing NCO methods can be categorized into two types: constructive heuristics (Khalil et al., 2017; Kool et al., 2018; Kwon et al., 2020; Hottung et al., 2020; Kim et al., 2022; Joshi et al., 2019; Fu et al., 2021; Geisler et al., 2021; Qiu et al., 2022; Sun & Yang, 2023; Luo et al., 2023; Vinyals et al., 2015; Bello et al., 2016; Nazari et al., 2018; Deudon et al., 2018; Xin et al., 2020; 2021; Kwon et al., 2021; Kim et al., 2021; Cheng et al., 2023; Drakulic et al., 2023) and improvement heuristics (Li et al., 2018; d O Costa et al., 2020; Wu et al., 2021; Chen & Tian, 2019; Li et al., 2023; Chen & Tian, 2019; Hottung et al., 2020; Joshi & Anand, 2022; Joshi et al., 2019) methods; the former can be further divided into two subtypes: autoregressive methods and non-autoregressive methods. The autoregressive methods grow a partial solution to a complete solution incrementally, and the non-autoregressive methods directly predict a heatmap. The improvement heuristic methods often work by iteratively improving a feasible initial solution.

**Coreset** Coreset is a popular data compression technique for clustering Chen (2009); Feldman & Langberg (2011); Braverman et al. (2022), regression Tukan et al. (2020) and optimization Huang et al. (2022); Wang et al. (2021). More relevant, Huang et al. Huang et al. (2021) proposed a sequential coreset for optimization problems with the Lipschitz smoothness property. Wang et al. (Wang et al., 2021) designed a coreset method for continuous-and-bounded learning (Shalev-Shwartz & Ben-David, 2014). However, these methods cannot offer an efficient method for aligning CO instances with training data in the inference phase.

**Optimal transportation (OT)** Discrete Wasserstein distance is a special case of OT, thus it can be computed by standard OT solvers. In recent years, a lot of algorithms have been proposed to solve OT problem. For example, interior point method can compute an $\epsilon_+$-approximation value for OT with $\widetilde{\mathcal{O}}(n^3)$ time in practice (Peyré et al., 2017) or $\widetilde{\mathcal{O}}(n^{2.5})$ in theory (Lee & Sidford, 2015). To obtain an $\epsilon_+$-approximation solution of OT, Sinkhorn algorithm takes $\widetilde{\mathcal{O}}(n^2/\epsilon_+^2)$ time (Dvurechensky et al., 2018; Lin et al., 2019) by solving the entropic regularization OT (Cuturi, 2013); the accelerated version of Sinkhorn algorithm yields $\widetilde{\mathcal{O}}(n^{2.5}/\epsilon_+)$ time (Guminov et al., 2020); especially, based on area-convexity and dual extrapolation, Jambulapati et al. (2019) achieved $\widetilde{\mathcal{O}}(n^2/\epsilon_+)$ time complexity.

## B OTHER ALGORITHMS

**Algorithm for computing RWD** We define two discrete probability measures

$$\alpha = \sum_{i=1}^{n} a_i \delta_{x_i}, \beta = \sum_{j=1}^{n} b_j \delta_{y_j} \in \mathcal{P}(\mathbb{R}^d), \tag{6}$$

where $\mathbf{a}, \mathbf{b} \in \mathbb{R}_+^n$ are their weights and $\{x_i\}_{i \in [n]}, \{y_j\}_{j \in [n]} \subseteq \mathbb{R}^d$ their locations; $\delta$ is the Dirac delta function. We denote the Wasserstein distance between $\alpha$ and $\beta$ by $W(\alpha, \beta)$.

The aim of Algorithm 3 is to find a rigid transformation $e$ such that $\mathcal{W}(\alpha, \beta) = W(e \circ \alpha, \beta)$. We initialize $\tilde{\alpha} = \alpha$. We solve $\mathcal{W}(\alpha, \beta)$ by updating the coupling $\mathbf{P}$ and rigid transformation $e$ alternatively. Specifically, we **first** obtain coupling $\mathbf{P}$ by computing $W(\tilde{\alpha}, \beta)$ according to the method in (Jambulapati et al., 2019). **Then**, we fix $\mathbf{P}$, and compute $\arg\min_e \mathcal{W}_{\mathbf{P}}(e \circ \alpha, \beta)$.

We partition the rigid transformation $e$ into translation transformation $e_1$ and orthogonal transformation $e_2$; that is, $e = e_2 \circ e_1$. The translation transformation can be updated as $e_1 = \sum_{i=1}^n \sum_{j=1}^n P_{ij} y_j - \sum_{i=1}^n \sum_{j=1}^n P_{ij} x_i$. For fixed $e_1, \mathbf{P}$, computing the optimal orthogonal transformation $e_2$ is an orthogonal Procrustes problem (Gower & Dijksterhuis, 2004). More specifically, we first obtain $\mathbf{M} = \sum_{ij} P_{ij} x_i y_j^T$, and apply singular value decomposition $\mathbf{M} = \mathbf{U} \mathbf{D} \mathbf{V}^{\mathbf{T}}$; then, we have

$$e_2 = \mathbf{U} \mathbf{V}^T. \tag{7}$$

---

**Algorithm 3** Algorithm for RWD

---

**Input:** $\alpha, \beta$

1: $t = 0, \tilde{\alpha} = \alpha$ ;
2: **for** $t < T_{align}$ **do**
3:     t = t +1;
4:     ▷ update coupling $\mathbf{P}$
      Obtain coupling $\mathbf{P}$ by computing $W(\tilde{\alpha}, \beta)$ according to the method in (Jambulapati et al., 2019);
5:     ▷ update translation transformation $e_1$ and orthogonal transformation $e_2$
      Compute $e_1 = \sum_{i=1}^n \sum_{j=1}^n P_{ij} y_j - \sum_{i=1}^n \sum_{j=1}^n P_{ij} x_i$;
      Compute $e_2 = \mathbf{U} \mathbf{V}^T$ according to Equation (7), and set $e = e_2 \circ e_1$;
      $\tilde{\alpha} = \sum_{i=1}^n a_i \delta_{(e \circ x_i)}$.
6: **end for**

**Output:** $\mathbf{P}, e$

---

**Time complexity of Algorithm 3**   We compute the RWD by alternating between optimizing the coupling matrix and the rigid transformation, which is a heuristic method. We assuming that the point dimension $d$ and the number of iterations $T_{align}$ are constants. For computing the coupling matrix, we solve an OT problem within $\tilde{O}(n^2)$ time (Jambulapati et al., 2019). The rigid transformation is obtained by solving an orthogonal Procrustes problem, which has a time complexity of $O(n^2 d + n d^2 + d^3)$. Thus, the overall complexity of this heuristic method remains $\tilde{O}(n^2)$.

**Algorithm for alignment**   Here, we introduce our alignment algorithm. We first initialize the current node $v$ as the root node $v_0$. Then, we walk from the root node to a leaf node by selecting the most similar measure with the test instance $\mu$.

---

**Algorithm 4** Algorithm for alignment

---

**Input:** the tree $\mathcal{T}$, CO instance $\mu$

1: $v = v_0$ ▷ The current node $v$ is initialized as root node $v_0$.
2: Initialize $target$ by any measure in coreset $\mathcal{S}$.
      ▷ $\mathcal{S}$ is the corresponding coreset maintained in tree $\mathcal{T}$.
3: Select the child $v'$ that is closest to $\mu$ under metric RWD.
4: Set $v = v'$.
5: **if** $\mathcal{W}(\mu, v'.\text{center}) \leq \mathcal{W}(\mu, target)$ **then**
6:     $target = v'.\text{center}$;
7: **end if**
8: If $v$ is not a leaf, jump to Line 3.

**Output:** $target$

---

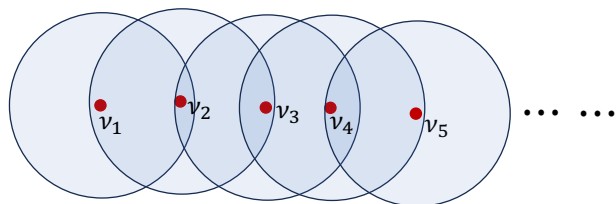

Figure 4: The illustration for error.

## C  OMITED PROOFS

*Proof of Theorem 3.3.* The proof sketch is listed here. First, we prove that by taking $(\mathcal{Q}, \mathsf{ddim}, r)$ as input, the coreset constructed by Algorithm 2 can cover the dataset $\mathcal{Q}$ by small balls of radius $r$ by using mathematical induction. Then, we obtain that the value difference of the loss function between the data itself and its representation is small by Lipschitz continuous property. Third, by setting some parameters properly, we turn the additive error into a relative error, and obtain an $\epsilon$-coreset $\mathcal{S}$.

Given a probability measure $\mu \in \mathcal{Q}$, we assume that its corresponding ball center in the $h$-th epoch is $\nu_h$.

**Claim C.1.** $\mathcal{W}^2(\mu, \nu_h) \leq h^2 \cdot \frac{r^2}{H^2}$.

*Proof.* The error is grown in a manner illustrated in Figure 4. Next, we prove this claim by using mathematical induction.
*Base Case*: For the case $h = 1$, in the 1-st epoch, we have $\mathcal{W}^2(\mu, \nu_1) \leq \frac{r^2}{H^2}$.

*Induction step:* For the case $h = m$, in the $m$-th epoch, we assume that

$$\mathcal{W}^2(\mu, \nu_m) \leq m^2 \cdot \frac{r^2}{H^2}. \tag{8}$$

Then, according to the generalized triangle inequalities in Lemma A.1, we have

$$\mathcal{W}^2(\mu, \nu_{m+1}) \leq (1 + t) \cdot \mathcal{W}^2(\mu, \nu_m) + (1 + \frac{1}{t}) \cdot \mathcal{W}^2(\nu_m, \nu_{m+1}). \tag{9}$$

Since the radius of small ball in Figure 4 is at most $\frac{r}{H}$, we have $\mathcal{W}^2(\nu_m, \nu_{m+1}) < \frac{r^2}{H^2}$. By using the the induction hypothesis and setting $t = \frac{1}{m}$, we have

$$\mathcal{W}^2(\mu, \nu_{m+1}) \leq (1 + \frac{1}{m}) \cdot m^2 \cdot \frac{r^2}{H^2} + (1 + m) \cdot \frac{r^2}{H^2} = (m + 1)^2 \frac{r^2}{H^2}. \tag{10}$$

Till now, we prove the case $h = m + 1$.

$\square$

Let $\nu := \nu_H$ be the representation of $\mu$ in Algorithm 2. According to Claim C.1, we have $\mathcal{W}^2(\mu, \nu) \leq r^2$.

Next, since loss function is $L$-Lipschitz continuous with respect to RWD on $\mathcal{Q}$, we have

$$|\ell(\mu, \theta) - \ell(\nu, \theta)| \leq L \cdot \mathcal{W}(\nu, \mu) \leq L \cdot r. \tag{11}$$

$$|\ell(\mathcal{Q}, \theta) - \ell(\mathcal{S}, \theta)| \leq \sum_{\mu \in \mathcal{Q}} w_{\mathcal{Q}}(\mu) \cdot |\ell(\mu, \theta) - \ell(\nu, \theta)| \leq \sum_{\mu \in \mathcal{Q}} w_{\mathcal{Q}}(\mu) \cdot L \cdot r = L \cdot r, \tag{12}$$

where $\nu \in \mathcal{S}$ is the corresponding representation of $\mu \in \mathcal{Q}$, and we have $w_{\mathcal{Q}}(\mathcal{Q}) := \sum_{\mu} w_{\mathcal{Q}}(\mu) = 1$ according to Definition 2.1.

Finally, we have $|\ell(\mathcal{Q}, \theta)| \geq \gamma$, by setting $r = \frac{\epsilon\gamma}{L}$, we obtain the coreset property

$$|\ell(\mathcal{Q}, \theta) - \ell(\mathcal{S}, \theta)| \leq \epsilon\gamma \leq \epsilon \cdot |\ell(\mathcal{Q}, \theta)| . \tag{13}$$

$\square$

# D  FULL EXPERIMENTS WITH TSP

We take TSP100 as an example to show the advantages of our coreset method. All experiments are conducted on an NVIDIA L20 GPU. We take the NCO Sun & Yang (2023) as our backbone solver, and set its training epoch as 20.

**Dataset**  We apply our method on TSP100-2D/3D Euclidean instances. The labels of TSP100-2D instances are obtained by using the LKH-3 heuristic solver (Helsgaun, 2017); each coordinate of the nodes in a TSP instance is generated by $x\%10$, where $x$ is randomly sampled either from a normal distribution $\mathcal{N}(0, \sigma^2)$ or uniform sampling $\mathcal{U}(0, 10)$. Our training data, TSP100-2D-$\mathcal{N}(0, 1)$ and TSP100-3D-$\mathcal{N}(0, 1)$, consists of 125,000 instances generated by the normal distribution $\mathcal{N}(0, 1)$ and 3000 instances by the uniform distribution $\mathcal{U}(0, 10)$. While the training dataset TSP100-2D-$\mathcal{U}(0, 10)$ consists of 128000 instances generated by uniform distribution $\mathcal{U}(0, 10)$.

Indeed, the distributions of the training dataset TSP100-2D-$\mathcal{N}(0, 1)$ and TSP100-3D-$\mathcal{N}(0, 1)$ are very close to the normal distribution $\mathcal{N}(0, 1)$. Hence, we regard the test data sampled from distribution $\mathcal{N}(0, 1)$ as having no distribution shift. The test data are sampled from a single distribution, either a normal distribution or uniform distribution. Specifically, we sample 1280 test data for TSP100, and 128 test data for other cases. Obviously, the uniform distribution has the highest entropy and thus the highest diversity. For these Gaussian distributions, the larger the variance, the larger the diversity.

As for the TSP100-3D dataset, we can directly extend the 2D instances to 3D instances by appending a third coordinate with a value of zero. Let $X = \{x_i\}_{i \in [n]} \subset \mathbb{R}^3$ are the nodes of an instance. We apply random rotation transformation $e$ on $X$; that is, $e(X) := \{e(x_i)\}_{i \in [n]}$. Intuitively, this TSP100-3D dataset has the same intrinsic complexity under metric RWD, which is the low doubling dimension in our assumption.

Table 4: Time statistics for different phases of training on TSP100-2D-$\mathcal{N}(0, 1)$.

| Method | Sample size | Labeling time | Coreset Time | Training Time | Total time |
|--------|-------------|---------------|--------------|---------------|------------|
| Org | 128000 | 4709 | - | 28563 | 33272 |
| US | 4003 | 147 | - | 1894 | 2041 |
| | 8245 | 304 | - | 2862 | 3166 |
| | 12951 | 475 | - | 4014 | 4489 |
| CS | 4003 | 145 | 691 | 1731 | 2567 |
| | 8245 | 305 | 1086 | 2747 | 4138 |
| | 12951 | 474 | 1283 | 3751 | 5508 |

**Setting**  We use the DIFUSCO (Sun & Yang, 2023) as our NCO solver. The model DIFUSCO with Greedy decoding solves TSP instances in an end-to-end manner. We set the learning rate as 0.0002 and the batch size as 64. The diffusion step is performed 50 times in inference phase. We use the `cosine` schedule described in (Sun & Yang, 2023). To quantify the performance of different methods, we use two criteria: the average tour length (Length) and the total runtime (Time). The term "Greedy" refers to the greedy decoding method of DIFUSCO, and "2-opt" is a post-processing used to improve solutions. The terms US, CS and CS-aligned represent uniform sampling, our coreset without alignment, and our coreset with alignment, respectively. We use the results on the full dataset with 128000 data as a baseline. In experiments, we first construct a coreset $\mathcal{S}$, and then take $|\mathcal{S}|$ samples by uniform sampling as training datasets.

**Results of TSP100-2D**  Tables 4 to 7 present the results on training dataset TSP100-2D-$\mathcal{N}(0, 1)$. The training datasets are generated by the uniform sampling and our coreset technique respectively.

Table 5: Comparison of uniform sampling and our coreset method using TSP100-2D-$\mathcal{N}(0,1)$ as the training dataset on test data TSP100-2D from different distributions.

| Sample size | Method | Test distribution | Greedy | | Greedy+2-opt | |
|---|---|---|---|---|---|---|
| | | | Length ($\downarrow$) | Time ($\downarrow$) | Length ($\downarrow$) | Time ($\downarrow$) |
| 128000 | Org | $\mathcal{N}(0,1)$ | 20.39 | 386 | 18.61 | 384 |
| | | $\mathcal{N}(0,2^2)$ | 42.41 | 381 | 37.47 | 387 |
| | | $\mathcal{N}(0,4^2)$ | 76.41 | 374 | 67.39 | 388 |
| | | $\mathcal{N}(0,8^2)$ | 87.18 | 379 | 77.86 | 388 |
| | | $\mathcal{U}(0,10)$ | 89.29 | 372 | 79.82 | 385 |
| 4003 | US | $\mathcal{N}(0,1)$ | 22.34 | 378 | 18.92 | 387 |
| | | $\mathcal{N}(0,2^2)$ | 51.59 | 376 | 38.25 | 388 |
| | | $\mathcal{N}(0,4^2)$ | 101.95 | 379 | 69.28 | 395 |
| | | $\mathcal{N}(0,8^2)$ | 118.83 | 379 | 80.38 | 402 |
| | | $\mathcal{U}(0,10)$ | 119.78 | 380 | 82.59 | 395 |
| | CS | $\mathcal{N}(0,1)$ | 22.21 | 372 | **18.87** | 379 |
| | | $\mathcal{N}(0,2^2)$ | **44.94** | 379 | **37.80** | 378 |
| | | $\mathcal{N}(0,4^2)$ | **80.63** | 372 | 67.92 | 379 |
| | | $\mathcal{N}(0,8^2)$ | 92.63 | 367 | 78.47 | 378 |
| | | $\mathcal{U}(0,10)$ | **94.73** | 373 | 80.64 | 377 |
| | CS-aligned | $\mathcal{N}(0,1)$ | **22.18** | 359 | 18.88 | 363 |
| | | $\mathcal{N}(0,2^2)$ | 45.00 | 357 | **37.80** | 362 |
| | | $\mathcal{N}(0,4^2)$ | 80.66 | 362 | **67.91** | 358 |
| | | $\mathcal{N}(0,8^2)$ | **92.59** | 362 | **78.41** | 358 |
| | | $\mathcal{U}(0,10)$ | 94.94 | 361 | **80.53** | 360 |
| 8245 | US | $\mathcal{N}(0,1)$ | 22.12 | 377 | 18.87 | 388 |
| | | $\mathcal{N}(0,2^2)$ | 45.59 | 381 | 37.86 | 389 |
| | | $\mathcal{N}(0,4^2)$ | 83.17 | 377 | 68.13 | 378 |
| | | $\mathcal{N}(0,8^2)$ | 95.16 | 380 | 78.81 | 385 |
| | | $\mathcal{U}(0,10)$ | 97.31 | 377 | 80.80 | 387 |
| | CS | $\mathcal{N}(0,1)$ | **21.79** | 366 | **18.84** | 383 |
| | | $\mathcal{N}(0,2^2)$ | **43.72** | 373 | **37.73** | 378 |
| | | $\mathcal{N}(0,4^2)$ | 78.72 | 372 | **67.79** | 378 |
| | | $\mathcal{N}(0,8^2)$ | **90.44** | 371 | 78.36 | 380 |
| | | $\mathcal{U}(0,10)$ | **92.99** | 374 | **80.35** | 377 |
| | CS-aligned | $\mathcal{N}(0,1)$ | 21.80 | 360 | 18.86 | 359 |
| | | $\mathcal{N}(0,2^2)$ | 43.77 | 354 | **37.73** | 356 |
| | | $\mathcal{N}(0,4^2)$ | **78.50** | 361 | 67.82 | 358 |
| | | $\mathcal{N}(0,8^2)$ | 90.54 | 350 | **78.32** | 359 |
| | | $\mathcal{U}(0,10)$ | 93.04 | 355 | 80.42 | 361 |
| 12951 | US | $\mathcal{N}(0,1)$ | 21.99 | 390 | 18.87 | 377 |
| | | $\mathcal{N}(0,2^2)$ | 44.77 | 376 | 37.81 | 384 |
| | | $\mathcal{N}(0,4^2)$ | 80.78 | 384 | 67.94 | 379 |
| | | $\mathcal{N}(0,8^2)$ | 93.16 | 373 | 78.52 | 381 |
| | | $\mathcal{U}(0,10)$ | 95.01 | 369 | 80.60 | 379 |
| | CS | $\mathcal{N}(0,1)$ | 21.57 | 372 | 18.81 | 382 |
| | | $\mathcal{N}(0,2^2)$ | **43.14** | 371 | **37.66** | 388 |
| | | $\mathcal{N}(0,4^2)$ | 77.80 | 369 | 67.58 | 379 |
| | | $\mathcal{N}(0,8^2)$ | 89.63 | 371 | **78.18** | 408 |
| | | $\mathcal{U}(0,10)$ | **92.01** | 378 | 80.23 | 375 |
| | CS-aligned | $\mathcal{N}(0,1)$ | **21.50** | 361 | **18.79** | 358 |
| | | $\mathcal{N}(0,2^2)$ | 43.18 | 361 | **37.66** | 364 |
| | | $\mathcal{N}(0,4^2)$ | **77.67** | 362 | **67.57** | 357 |
| | | $\mathcal{N}(0,8^2)$ | **89.60** | 357 | **78.18** | 361 |
| | | $\mathcal{U}(0,10)$ | **92.01** | 358 | **80.21** | 359 |

The results show that both our method and uniform sampling method perform better as the sample size increases. Meanwhile, as the sample size decreases, the advantage of our methods compared to uniform sampling becomes increasingly evident.

Moreover, Tables 5 and 6 demonstrate that our method is robust to distribution shift. Specifically, the training data is sampled from a normal distribution $\mathcal{N}(0, 1)$, while the test data is sampled from normal distributions $\mathcal{N}(0, 1)$, $\mathcal{N}(0, 4^2)$ and a uniform distribution $\mathcal{U}(0, 10)$. The test distributions $\mathcal{N}(0, 4^2)$ and $\mathcal{U}(0, 10)$ are significantly different from the training distribution $\mathcal{N}(0, 1)$, which represent substantial distribution shifts. The results in Tables 5 and 6 show that our method consistently outperforms the baselines, demonstrating its robustness to distribution shifts.

Furthermore, Table 6 shows that models trained on our coreset can generalize better to larger problem sizes such as TSP200, TSP500 and TSP1000. Moreover, Table 7 confirms that our method outperforms other approaches on the TSPLIB dataset. Table 4 shows the time efficiency of our coreset method.

Table 6: Comparison of uniform sampling and our coreset method using TSP100-2D-$\mathcal{N}(0, 1)$ as the training dataset on test data of varying sizes. We fix the sample size as 12951.

| TSP size | Method | Test distribution | Greedy | | Greedy+2-opt | |
|---|---|---|---|---|---|---|
| | | | Length ($\downarrow$) | Time ($\downarrow$) | Length ($\downarrow$) | Time ($\downarrow$) |
| TSP-200 | Org | $\mathcal{N}(0, 1)$ | 29.85 | 107 | 26.61 | 111 |
| | | $\mathcal{N}(0, 2^2)$ | 60.87 | 109 | 53.38 | 110 |
| | | $\mathcal{N}(0, 4^2)$ | 109.70 | 110 | 94.90 | 111 |
| | | $\mathcal{N}(0, 8^2)$ | 123.39 | 110 | 108.41 | 110 |
| | | $\mathcal{U}(0, 10)$ | 126.99 | 107 | 111.50 | 111 |
| TSP-500 | Org | $\mathcal{N}(0, 1)$ | 50.55 | 1012 | 42.38 | 1018 |
| | | $\mathcal{N}(0, 2^2)$ | 102.65 | 1012 | 84.89 | 1018 |
| | | $\mathcal{N}(0, 4^2)$ | 184.40 | 1012 | 150.47 | 1018 |
| | | $\mathcal{N}(0, 8^2)$ | 204.53 | 1010 | 171.14 | 1014 |
| | | $\mathcal{U}(0, 10)$ | 210.70 | 1012 | 175.41 | 1016 |
| TSP-1000 | Org | $\mathcal{N}(0, 1)$ | 77.15 | 2826 | 60.56 | 2840 |
| | | $\mathcal{N}(0, 2^2)$ | 157.12 | 2826 | 121.45 | 2839 |
| | | $\mathcal{N}(0, 4^2)$ | 281.64 | 4224 | 214.55 | 2833 |
| | | $\mathcal{N}(0, 8^2)$ | 310.94 | 4218 | 243.77 | 4242 |
| | | $\mathcal{U}(0, 10)$ | 317.55 | 4218 | 249.28 | 4236 |
| TSP-200 | US | $\mathcal{N}(0, 1)$ | 33.69 | 109 | 27.14 | 112 |
| | | $\mathcal{N}(0, 2^2)$ | 69.77 | 109 | 54.28 | 112 |
| | | $\mathcal{N}(0, 4^2)$ | 125.99 | 108 | 96.70 | 112 |
| | | $\mathcal{N}(0, 8^2)$ | 143.76 | 109 | 110.64 | 113 |
| | | $\mathcal{U}(0, 10)$ | 145.41 | 109 | 113.39 | 112 |
| | CS | $\mathcal{N}(0, 1)$ | **30.75** | 107 | 26.69 | 110 |
| | | $\mathcal{N}(0, 2^2)$ | 62.08 | 109 | **53.36** | 112 |
| | | $\mathcal{N}(0, 4^2)$ | **110.48** | 109 | 94.84 | 111 |
| | | $\mathcal{N}(0, 8^2)$ | 127.06 | 107 | 108.76 | 111 |
| | | $\mathcal{U}(0, 10)$ | 129.77 | 107 | **111.47** | 109 |
| | CS-aligned | $\mathcal{N}(0, 1)$ | 30.77 | 77 | **26.68** | 79 |
| | | $\mathcal{N}(0, 2^2)$ | **61.87** | 76 | 53.38 | 79 |
| | | $\mathcal{N}(0, 4^2)$ | 110.99 | 78 | **94.59** | 79 |
| | | $\mathcal{N}(0, 8^2)$ | **126.69** | 76 | **108.42** | 79 |
| | | $\mathcal{U}(0, 10)$ | **129.28** | 76 | 111.49 | 78 |
| | US | $\mathcal{N}(0, 1)$ | 59.81 | 1012 | 43.41 | 1020 |
| | | $\mathcal{N}(0, 2^2)$ | 126.00 | 1013 | 86.76 | 1022 |
| | | $\mathcal{N}(0, 4^2)$ | 237.72 | 1012 | 154.28 | 1022 |
| | | $\mathcal{N}(0, 8^2)$ | 261.79 | 1013 | 176.13 | 1022 |
| | | $\mathcal{U}(0, 10)$ | 263.66 | 1015 | 180.75 | 1022 |

Continued on next page

TSP-500

**– continued from previous page**

| TSP size | Method | Test distribution | Greedy | | Greedy+2-opt | |
|---|---|---|---|---|---|---|
| | | | Length ($\downarrow$) | Time ($\downarrow$) | Length ($\downarrow$) | Time ($\downarrow$) |
| | CS | $\mathcal{N}(0,1)$ | **49.11** | 1012 | **42.25** | 1016 |
| | | $\mathcal{N}(0,2^2)$ | **99.58** | 1012 | 84.60 | 1019 |
| | | $\mathcal{N}(0,4^2)$ | **178.56** | 1010 | **149.50** | 1016 |
| | | $\mathcal{N}(0,8^2)$ | 205.86 | 1012 | **170.54** | 1016 |
| | | $\mathcal{U}(0,10)$ | **208.36** | 1011 | **174.93** | 1016 |
| | CS-aligned | $\mathcal{N}(0,1)$ | 49.38 | 680 | 42.26 | 683 |
| | | $\mathcal{N}(0,2^2)$ | 99.66 | 680 | **84.52** | 684 |
| | | $\mathcal{N}(0,4^2)$ | 178.88 | 679 | 149.63 | 682 |
| | | $\mathcal{N}(0,8^2)$ | **204.42** | 678 | 170.62 | 682 |
| | | $\mathcal{U}(0,10)$ | 208.77 | 678 | 175.03 | 682 |
| | US | $\mathcal{N}(0,1)$ | 94.71 | 2823 | 61.72 | 2848 |
| | | $\mathcal{N}(0,2^2)$ | 199.22 | 2825 | 123.56 | 2849 |
| | | $\mathcal{N}(0,4^2)$ | 382.77 | 4224 | 219.16 | 2847 |
| | | $\mathcal{N}(0,8^2)$ | 421.29 | 4218 | 249.08 | 4258 |
| | | $\mathcal{U}(0,10)$ | 426.61 | 4215 | 255.95 | 4254 |
| TSP-1000 | CS | $\mathcal{N}(0,1)$ | **69.76** | 2823 | 59.59 | 2833 |
| | | $\mathcal{N}(0,2^2)$ | 141.06 | 2825 | 119.64 | 2832 |
| | | $\mathcal{N}(0,4^2)$ | **252.92** | 4224 | **210.71** | 2832 |
| | | $\mathcal{N}(0,8^2)$ | **293.49** | 4218 | 240.42 | 4236 |
| | | $\mathcal{U}(0,10)$ | **299.80** | 4215 | **246.57** | 4234 |
| | CS-aligned | $\mathcal{N}(0,1)$ | 69.96 | 2825 | **59.57** | 2832 |
| | | $\mathcal{N}(0,2^2)$ | **140.95** | 2822 | **119.39** | 2833 |
| | | $\mathcal{N}(0,4^2)$ | 253.63 | 2821 | 210.81 | 2830 |
| | | $\mathcal{N}(0,8^2)$ | 293.82 | 2815 | **240.11** | 2826 |
| | | $\mathcal{U}(0,10)$ | 300.03 | 2812 | 246.61 | 2827 |

Table 7: Comparison of uniform sampling and our coreset method using TSP100-2D-$\mathcal{N}(0,1)$ as the training dataset on test data TSPLIB(Reinelt, 1991).

| Sample size | Method | Test distribution | Greedy | | Greedy+2-opt | |
|---|---|---|---|---|---|---|
| | | | Length ($\downarrow$) | Time ($\downarrow$) | Length ($\downarrow$) | Time ($\downarrow$) |
| 128000 | Org | $\mathcal{N}(0,1)$ | 129.35 | 108 | 112.23 | 106 |
| 4003 | US | $\mathcal{N}(0,1)$ | 190.79 | 109 | 115.87 | 108 |
| | CS | $\mathcal{N}(0,1)$ | 153.08 | 107 | **113.56** | 108 |
| | CS-aligned | $\mathcal{N}(0,1)$ | **152.70** | 103 | 113.71 | 105 |
| 8245 | US | $\mathcal{N}(0,1)$ | 166.40 | 106 | 114.47 | 108 |
| | CS | $\mathcal{N}(0,1)$ | 140.49 | 107 | 113.04 | 106 |
| | CS-aligned | $\mathcal{N}(0,1)$ | **140.18** | 104 | **112.91** | 104 |
| 12951 | US | $\mathcal{N}(0,1)$ | 162.19 | 107 | 114.31 | 107 |
| | CS | $\mathcal{N}(0,1)$ | 133.63 | 106 | **112.45** | 105 |
| | CS-aligned | $\mathcal{N}(0,1)$ | **133.14** | 103 | 112.52 | 110 |

**Results of TSP100-3D-$\mathcal{N}(0,1)$**   Tables 8 to 10 show the results of TSP100-3D-$\mathcal{N}(0,1)$. Table 8 illustrates the overall acceleration improvement. For the test data without distribution shift, our method has comparable performance; for the test data occurring distribution shift, our method has better performance. Moreover, our coreset with alignment version performs better in the TSP-3D case. From Tables 9 and 10, alignment can perform better for higher dimension dataset (i.e., TSP-3D). Thus, the alignment version is promising for tackling high-dimensional data.

Table 8: Time statistics for different phases of training on TSP100-3D-$\mathcal{N}(0,1)$.

| Method | Sample size | Labeling time | Coreset Time | Training Time | Total time |
|--------|-------------|---------------|--------------|---------------|------------|
| Org | 128000 | 4940 | - | 30671 | 35611 |
| US | 4103 | 160 | - | 2102 | 2262 |
| | 7960 | 307 | - | 2729 | 3036 |
| | 12058 | 466 | - | 3514 | 3980 |
| CS | 4103 | 159 | 1012 | 2080 | 1379 |
| | 7960 | 309 | 1177 | 2712 | 4198 |
| | 12058 | 463 | 1675 | 3493 | 5631 |

Table 10: Comparison of uniform sampling and our coreset method with training dataset TSP100-3D-$\mathcal{N}(0,1)$ on test data of varying sizes. We fix the sample size as 12058.

| TSP size | Method | Test distribution | Greedy Length ($\downarrow$) | Time ($\downarrow$) | Alignment time ($\downarrow$) |
|----------|--------|-------------------|-------|------|----------------|
| TSP-200 | Org | $\mathcal{N}(0,1)$ | 30.02 | 77 | - |
| | | $\mathcal{N}(0,2^2)$ | 61.03 | 76 | - |
| | | $\mathcal{N}(0,4^2)$ | 109.78 | 77 | - |
| | | $\mathcal{N}(0,8^2)$ | 126.15 | 76 | - |
| | | $\mathcal{U}(0,10)$ | 128.35 | 77 | - |
| TSP-500 | Org | $\mathcal{N}(0,1)$ | 48.66 | 682 | - |
| | | $\mathcal{N}(0,2^2)$ | 101.71 | 682 | - |
| | | $\mathcal{N}(0,4^2)$ | 184.94 | 682 | - |
| | | $\mathcal{N}(0,8^2)$ | 210.36 | 680 | - |
| | | $\mathcal{U}(0,10)$ | 212.62 | 683 | - |
| TSP-1000 | Org | $\mathcal{N}(0,1)$ | 69.08 | 2828 | - |
| | | $\mathcal{N}(0,2^2)$ | 144.10 | 2826 | - |
| | | $\mathcal{N}(0,4^2)$ | 264.58 | 2824 | - |
| | | $\mathcal{N}(0,8^2)$ | 303.02 | 2821 | - |
| | | $\mathcal{U}(0,10)$ | 313.23 | 2818 | - |
| TSP-200 | US | $\mathcal{N}(0,1)$ | **32.30** | 76 | - |
| | | $\mathcal{N}(0,2^2)$ | 67.81 | 77 | - |
| | | $\mathcal{N}(0,4^2)$ | 130.76 | 77 | - |
| | | $\mathcal{N}(0,8^2)$ | 152.74 | 77 | - |
| | | $\mathcal{U}(0,10)$ | 153.90 | 77 | - |
| | CS | $\mathcal{N}(0,1)$ | 32.72 | 76 | - |
| | | $\mathcal{N}(0,2^2)$ | **66.08** | 76 | - |
| | | $\mathcal{N}(0,4^2)$ | 120.64 | 78 | - |
| | | $\mathcal{N}(0,8^2)$ | 139.91 | 77 | - |
| | | $\mathcal{U}(0,10)$ | 142.84 | 77 | - |
| | CS-aligned | $\mathcal{N}(0,1)$ | 33.25 | 76 | 2 |
| | | $\mathcal{N}(0,2^2)$ | 69.79 | 75 | 5 |
| | | $\mathcal{N}(0,4^2)$ | **118.44** | 77 | 7 |
| | | $\mathcal{N}(0,8^2)$ | **135.90** | 77 | 10 |
| | | $\mathcal{U}(0,10)$ | **138.27** | 76 | 12 |
| | US | $\mathcal{N}(0,1)$ | 58.45 | 682 | - |
| | | $\mathcal{N}(0,2^2)$ | 127.53 | 681 | - |
| | | $\mathcal{N}(0,4^2)$ | 264.93 | 682 | - |
| | | $\mathcal{N}(0,8^2)$ | 316.73 | 681 | - |
| | | $\mathcal{U}(0,10)$ | 318.39 | 680 | - |
| TSP-500 | CS | $\mathcal{N}(0,1)$ | 59.69 | 682 | - |

**– continued from previous page**

| TSP size | Method | Test distribution | Greedy Length (↓) | Time (↓) | |
|---|---|---|---|---|---|
| | | $\mathcal{N}(0, 2^2)$ | 122.24 | 681 | - |
| | | $\mathcal{N}(0, 4^2)$ | 231.96 | 682 | - |
| | | $\mathcal{N}(0, 8^2)$ | 266.08 | 677 | - |
| | | $\mathcal{U}(0, 10)$ | 268.75 | 680 | - |
| | | $\mathcal{N}(0, 1)$ | **56.47** | 682 | 16 |
| | | $\mathcal{N}(0, 2^2)$ | **121.33** | 681 | 19 |
| | CS-aligned | $\mathcal{N}(0, 4^2)$ | **205.45** | 682 | 23 |
| | | $\mathcal{N}(0, 8^2)$ | **238.14** | 681 | 26 |
| | | $\mathcal{U}(0, 10)$ | **245.01** | 681 | 30 |
| | | $\mathcal{N}(0, 1)$ | **86.01** | 2828 | - |
| | | $\mathcal{N}(0, 2^2)$ | 191.39 | 2828 | - |
| | US | $\mathcal{N}(0, 4^2)$ | 397.40 | 2825 | - |
| | | $\mathcal{N}(0, 8^2)$ | 470.55 | 2820 | - |
| | | $\mathcal{U}(0, 10)$ | 550.11 | 2819 | - |
| | | $\mathcal{N}(0, 1)$ | 86.57 | 2826 | - |
| | | $\mathcal{N}(0, 2^2)$ | **181.84** | 2827 | - |
| TSP-1000 | CS | $\mathcal{N}(0, 4^2)$ | 342.64 | 2822 | - |
| | | $\mathcal{N}(0, 8^2)$ | 395.66 | 2820 | - |
| | | $\mathcal{U}(0, 10)$ | 441.82 | 2818 | - |
| | | $\mathcal{N}(0, 1)$ | 86.87 | 2828 | 236 |
| | | $\mathcal{N}(0, 2^2)$ | 186.88 | 2823 | 433 |
| | CS-aligned | $\mathcal{N}(0, 4^2)$ | **320.68** | 2825 | 624 |
| | | $\mathcal{N}(0, 8^2)$ | **372.34** | 2820 | 861 |
| | | $\mathcal{U}(0, 10)$ | **383.92** | 2819 | 980 |

Then, we take TSP-3D as an example to compare the performance of the random heuristic alignment method (CS-rand-aligned) with our proposed alignment method (CS-aligned).

As shown in Table 11, the results demonstrate that the CS-rand-aligned method performs similarly to the unaligned approach, providing little improvement. In contrast, our alignment method significantly enhances performance, confirming its practical effectiveness.

Table 11: Comparison of rand alignment (CS-rand-aligned) and our alignment (CS-aligned) method with training dataset TSP100-3D-$\mathcal{N}(0, 1)$ on test data of varying sizes. We fix the sample size as 12058.

| TSP size | Method | Test distribution | Greedy Length (↓) | Time (↓) | Alignment time (↓) |
|---|---|---|---|---|---|
| | US | $\mathcal{U}(0, 10)$ | 100.41 | 37 | - |
| TSP-100 | CS | $\mathcal{U}(0, 10)$ | 95.27 | 36 | - |
| | CS-aligned | $\mathcal{U}(0, 10)$ | **94.13** | 37 | 3 |
| | CS-rand-aligned | $\mathcal{U}(0, 10)$ | 95.62 | 37 | 11 |
| | US | $\mathcal{U}(0, 10)$ | 153.90 | 77 | - |
| TSP-200 | CS | $\mathcal{U}(0, 10)$ | 142.84 | 77 | - |
| | CS-aligned | $\mathcal{U}(0, 10)$ | **138.27** | 76 | 12 |
| | CS-rand-aligned | $\mathcal{U}(0, 10)$ | 144.84 | 75 | 17 |
| | US | $\mathcal{U}(0, 10)$ | 318.39 | 680 | - |
| TSP-500 | CS | $\mathcal{U}(0, 10)$ | 268.75 | 680 | - |
| | CS-aligned | $\mathcal{U}(0, 10)$ | **245.01** | 681 | 30 |
| | CS-rand-aligned | $\mathcal{U}(0, 10)$ | 255.92 | 674 | 23 |
| | US | $\mathcal{U}(0, 10)$ | 550.11 | 2819 | - |
| TSP-1000 | CS | $\mathcal{U}(0, 10)$ | 441.82 | 2826 | - |
| | CS-aligned | $\mathcal{U}(0, 10)$ | **383.92** | 2818 | 980 |

Continued on next page

| TSP size | Method | Test distribution | Greedy | | |
| | | | Length ($\downarrow$) | Time ($\downarrow$) | |
| --- | --- | --- | --- | --- | --- |
| | CS-rand-aligned | $\mathcal{U}(0, 10)$ | 429.90 | 2817 | 493 |

**– continued from previous page**

**Results of TSP100-2D-$\mathcal{U}(0, 10)$**   Tables 12 to 15 present the results on training dataset TSP100-2D-$\mathcal{U}(0, 10)$. The training datasets are generated by the uniform sampling and our coreset technique respectively. Table 12 clearly demonstrates the overall acceleration improvement. All the results show that both our method and uniform sampling method perform better as the sample size increases. For both the test data with distribution shift and without distribution shift, our method consistently shows better performance. Table 14 further illustrates that our method can generalize to large-scale TSP problems better. Table 15 confirms that our method outperforms other approaches on the TSPLIB dataset.

**Results of TSP100-3D-$\mathcal{U}(0, 10)$**   Tables 16 and 17 present the results on training dataset TSP100-2D-$\mathcal{U}(0, 10)$.

## E   FULL EXPERIMENTS WITH MIS

**Dataset**   For Maximal Independent Set (MIS), we train our model on ER-[90-100] dataset (Erd6s & Rényi, 1960), where ER-[$n$-$N$] means that the graph contains $n$ to $N$ nodes. We set the connection probability as 0.15 as in (Sun & Yang, 2023). The labels of our MIS datasets are obtained by using the KaMIS2 solver. Our training dataset of MIS consists of 128000 ER-[90-100] instances. We evaluate our method on ER-[90-100], ER-[400-500], ER-[900-1000] and SATLIB.

**Setting**   We use the DIFUSCO (Sun & Yang, 2023) as our NCO solver. To quantify the performance of different methods, we use two criteria: the average size of the independent set (Size) and the total runtime (Time). The term "Greedy" refers to the greedy decoding method of DIFUSCO, and "Sampling" is the sampling decoding. The terms US and CS represent uniform sampling and our coreset method, respectively. We use the results on the full dataset with 128000 data as a baseline.

**Results of MIS**   Tables 18, 20 and 27 present the results on training dataset ER-[90-100]. The training datasets are generated by the uniform sampling and our coreset technique respectively. Table 18 clearly demonstrates the overall acceleration improvement. All the results show that both our method and uniform sampling method perform better as the sample size increases. Table 27 demonstrates that our method can generalize to large-scale ER problems better. Table 20 show that the performance of our method and uniform sampling method is similar on SATLIB.

## F   FULL EXPERIMENTS WITH CVRP

In this section, we take CVRP100 as an example to show the advantages of our coreset method. All experiments of CVRP are conducted on NVIDIA GeForce RTX 3090 GPU. We take the NCO (Luo et al., 2023) as our backbone solver, and set its training epoch as 20.

**Dataset**   We apply our method on CVRP100 Euclidean instances. The labels of CVRP100 instances are obtained by using the LKH-3 heuristic solver (Helsgaun, 2017); each coordinate of the nodes in a CVRP instance is generated by $x\%1$, where $x$ is randomly sampled either from a normal distribution $\mathcal{N}(0, \sigma^2)$ or uniform sampling $\mathcal{U}(0, 1)$. Our training data CVRP100-$\mathcal{N}(0, 0.1^2)$ consists of 125,000 instances generated by the normal distribution $\mathcal{N}(0, 0.1^2)$ and 3000 instances by the uniform distribution $\mathcal{U}(0, 1)$. While the training dataset CVRP100-$\mathcal{U}(0, 1)$ consists of 128000 instances generated by uniform distribution $\mathcal{U}(0, 1)$.

Indeed, the distributions of the training dataset CVRP100-$\mathcal{N}(0, 0.1^2)$ is very close to the normal distribution $\mathcal{N}(0, 0.1^2)$. Hence, we regard the test data sampled from distribution $\mathcal{N}(0, 0.1^2)$ as having no distribution shift. The test data are sampled from a single distribution, either a normal distribution or uniform distribution. Specifically, we sample 1280 test data for CVRP100, and 128

Table 9: Comparison of uniform sampling and our coreset method using TSP100-3D-$\mathcal{N}(0,1)$ as the training dataset on test data TSP100-3D from different distributions.

| Sample size | Method | Test distribution | Greedy Length ($\downarrow$) | Time ($\downarrow$) | Alignment time ($\downarrow$) |
|---|---|---|---|---|---|
| 128000 | Org | $\mathcal{N}(0,1)$ | 20.80 | 364 | - |
| | | $\mathcal{N}(0,2^2)$ | 42.25 | 366 | - |
| | | $\mathcal{N}(0,4^2)$ | 76.57 | 362 | - |
| | | $\mathcal{N}(0,8^2)$ | 88.45 | 360 | - |
| | | $\mathcal{U}(0,10)$ | 90.55 | 358 | - |
| 4103 | US | $\mathcal{N}(0,1)$ | 24.92 | 480 | - |
| | | $\mathcal{N}(0,2^2)$ | 49.96 | 482 | - |
| | | $\mathcal{N}(0,4^2)$ | 96.60 | 483 | - |
| | | $\mathcal{N}(0,8^2)$ | 116.65 | 482 | - |
| | | $\mathcal{U}(0,10)$ | 119.78 | 481 | - |
| | CS | $\mathcal{N}(0,1)$ | 24.89 | 364 | - |
| | | $\mathcal{N}(0,2^2)$ | 50.46 | 384 | - |
| | | $\mathcal{N}(0,4^2)$ | 106.35 | 360 | - |
| | | $\mathcal{N}(0,8^2)$ | 109.12 | 360 | - |
| | | $\mathcal{U}(0,10)$ | 111.63 | 353 | - |
| | CS-aligned | $\mathcal{N}(0,1)$ | **23.36** | 479 | 2 |
| | | $\mathcal{N}(0,2^2)$ | **47.00** | 480 | 4 |
| | | $\mathcal{N}(0,4^2)$ | **91.94** | 479 | 7 |
| | | $\mathcal{N}(0,8^2)$ | **106.50** | 482 | 9 |
| | | $\mathcal{U}(0,10)$ | **108.81** | 483 | 11 |
| 7960 | US | $\mathcal{N}(0,1)$ | 23.62 | 477 | - |
| | | $\mathcal{N}(0,2^2)$ | 48.32 | 477 | - |
| | | $\mathcal{N}(0,4^2)$ | 92.92 | 484 | - |
| | | $\mathcal{N}(0,8^2)$ | 111.98 | 479 | - |
| | | $\mathcal{U}(0,10)$ | 115.62 | 481 | - |
| | CS | $\mathcal{N}(0,1)$ | 23.41 | 362 | - |
| | | $\mathcal{N}(0,2^2)$ | 47.10 | 361 | - |
| | | $\mathcal{N}(0,4^2)$ | 86.20 | 362 | - |
| | | $\mathcal{N}(0,8^2)$ | 99.50 | 365 | - |
| | | $\mathcal{U}(0,10)$ | 101.25 | 359 | - |
| | CS-aligned | $\mathcal{N}(0,1)$ | **22.83** | 476 | 12 |
| | | $\mathcal{N}(0,2^2)$ | **46.06** | 474 | 14 |
| | | $\mathcal{N}(0,4^2)$ | **85.71** | 483 | 16 |
| | | $\mathcal{N}(0,8^2)$ | **97.61** | 480 | 17 |
| | | $\mathcal{U}(0,10)$ | **99.10** | 481 | 19 |
| 12058 | US | $\mathcal{N}(0,1)$ | **22.10** | 360 | - |
| | | $\mathcal{N}(0,2^2)$ | 45.52 | 368 | - |
| | | $\mathcal{N}(0,4^2)$ | 84.28 | 368 | - |
| | | $\mathcal{N}(0,8^2)$ | 98.57 | 372 | - |
| | | $\mathcal{U}(0,10)$ | 100.40 | 367 | - |
| | CS | $\mathcal{N}(0,1)$ | **22.10** | 371 | - |
| | | $\mathcal{N}(0,2^2)$ | **44.40** | 362 | - |
| | | $\mathcal{N}(0,4^2)$ | 80.47 | 361 | - |
| | | $\mathcal{N}(0,8^2)$ | 93.00 | 361 | - |
| | | $\mathcal{U}(0,10)$ | 95.25 | 362 | - |
| | CS-aligned | $\mathcal{N}(0,1)$ | 22.30 | 396 | 21 |
| | | $\mathcal{N}(0,2^2)$ | 44.80 | 371 | 22 |
| | | $\mathcal{N}(0,4^2)$ | **79.96** | 388 | 23 |
| | | $\mathcal{N}(0,8^2)$ | **92.28** | 377 | 25 |
| | | $\mathcal{U}(0,10)$ | **94.13** | 366 | 26 |

Table 12: Time statistics for different phases of training on TSP100-2D-$\mathcal{U}(0, 10)$.

| Method | Sample size | Labeling time | Coreset Time | Training Time | Total time |
|--------|-------------|---------------|--------------|---------------|------------|
| Org | 128000 | 4883 | - | 27686 | 32569 |
| US | 4003 | 153 | - | 2387 | 2540 |
| | 8245 | 315 | - | 3217 | 3532 |
| | 12951 | 495 | - | 3578 | 4073 |
| CS | 4003 | 154 | 859 | 2331 | 3344 |
| | 8245 | 315 | 841 | 3217 | 4373 |
| | 12951 | 493 | 1293 | 3516 | 5302 |

Table 13: Comparison of uniform sampling and our coreset method using TSP100-2D-$\mathcal{U}(0, 10)$ as the training dataset on test data TSP100-2D from different distributions.

| Sample size | Method | Test distribution | Greedy | | Greedy+2-opt | |
|-------------|--------|-------------------|--------|------|--------------|------|
| | | | Length ($\downarrow$) | Time ($\downarrow$) | Length ($\downarrow$) | Time ($\downarrow$) |
| 128000 | Org | $\mathcal{U}(0, 10)$ | 86.77 | 358 | 79.53 | 353 |
| 4003 | US | $\mathcal{U}(0, 10)$ | 100.62 | 355 | 80.97 | 375 |
| | CS | $\mathcal{U}(0, 10)$ | **97.69** | 461 | **80.94** | 383 |
| | CS-aligned | $\mathcal{U}(0, 10)$ | 97.71 | 363 | 80.95 | 369 |
| 8245 | US | $\mathcal{U}(0, 10)$ | 92.66 | 352 | **80.33** | 365 |
| | CS | $\mathcal{U}(0, 10)$ | **93.12** | 381 | 80.38 | 372 |
| | CS-aligned | $\mathcal{U}(0, 10)$ | 93.27 | 351 | 80.41 | 354 |
| 12951 | US | $\mathcal{U}(0, 10)$ | 91.98 | 362 | 80.34 | 367 |
| | CS | $\mathcal{U}(0, 10)$ | **92.33** | 354 | **80.23** | 356 |
| | CS-aligned | $\mathcal{U}(0, 10)$ | 92.39 | 360 | 80.28 | 367 |

Table 14: Comparison of uniform sampling and our coreset method using TSP100-2D-$\mathcal{U}(0, 10)$ as the training dataset on test data of varying sizes. We fix the sample size as 12951.

| TSP size | Method | Test distribution | Greedy | | Greedy+2-opt | |
|----------|--------|-------------------|--------|------|--------------|------|
| | | | Length ($\downarrow$) | Time ($\downarrow$) | Length ($\downarrow$) | Time ($\downarrow$) |
| TSP-200 | Org | $\mathcal{U}(0, 10)$ | 125.06 | 72 | 110.89 | 73 |
| TSP-500 | Org | $\mathcal{U}(0, 10)$ | 195.68 | 675 | 172.19 | 677 |
| TSP-1000 | Org | $\mathcal{U}(0, 10)$ | 276.69 | 2815 | 241.59 | 2824 |
| TSP-200 | US | $\mathcal{U}(0, 10)$ | 130.05 | 76 | 111.71 | 81 |
| | CS | $\mathcal{U}(0, 10)$ | **129.52** | 75 | **111.31** | 81 |
| | CS-aligned | $\mathcal{U}(0, 10)$ | 129.95 | 77 | 111.35 | 76 |
| TSP-500 | US | $\mathcal{U}(0, 10)$ | 238.46 | 677 | 177.30 | 682 |
| | CS | $\mathcal{U}(0, 10)$ | 221.76 | 677 | 175.94 | 682 |
| | CS-aligned | $\mathcal{U}(0, 10)$ | **221.72** | 675 | **175.89** | 683 |
| TSP-1000 | US | $\mathcal{U}(0, 10)$ | 434.16 | 2811 | 254.37 | 2837 |
| | CS | $\mathcal{U}(0, 10)$ | **346.71** | 2811 | 251.40 | 2831 |
| | CS-aligned | $\mathcal{U}(0, 10)$ | 347.46 | 2813 | **251.35** | 2833 |

Table 15: Comparison of uniform sampling and our coreset method using TSP100-2D-$\mathcal{U}(0, 10)$ as the training dataset on test data TSPLIB(Reinelt, 1991).

| Sample size | Method | Test distribution | Greedy | | Greedy+2-opt | |
|---|---|---|---|---|---|---|
| | | | Length (↓) | Time (↓) | Length (↓) | Time (↓) |
| 128000 | Org | $\mathcal{N}(0, 1)$ | 125.39 | 102 | 110.90 | 104 |
| 4003 | US | $\mathcal{U}(0, 10)$ | 183.71 | 104 | 115.21 | 110 |
| | CS | $\mathcal{U}(0, 10)$ | **164.81** | 106 | 114.50 | 108 |
| | CS-aligned | $\mathcal{U}(0, 10)$ | 166.04 | 105 | **114.41** | 103 |
| 8245 | US | $\mathcal{U}(0, 10)$ | 161.75 | 107 | 113.69 | 107 |
| | CS | $\mathcal{U}(0, 10)$ | **132.06** | 103 | **111.52** | 111 |
| | CS-aligned | $\mathcal{U}(0, 10)$ | 145.90 | 103 | 112.86 | 103 |
| 12951 | US | $\mathcal{U}(0, 10)$ | 157.10 | 104 | 113.98 | 105 |
| | CS | $\mathcal{U}(0, 10)$ | **140.97** | 104 | 112.96 | 106 |
| | CS-aligned | $\mathcal{U}(0, 10)$ | 141.81 | 501 | **112.81** | 110 |

Table 16: Comparison of uniform sampling and our coreset method using TSP100-3D-$\mathcal{U}(0, 10)$ as the training dataset on test data TSP100-3D from different distributions.

| Sample size | Method | Test distribution | Greedy | |
|---|---|---|---|---|
| | | | Length (↓) | Time (↓) |
| 128000 | Org | $\mathcal{U}(0, 10)$ | 87.84 | 350 |
| 4122 | US | $\mathcal{U}(0, 10)$ | 108.74 | 350 |
| | CS | $\mathcal{U}(0, 10)$ | 108.45 | 352 |
| | CS-aligned | $\mathcal{U}(0, 10)$ | **106.11** | 381 |
| 8245 | US | $\mathcal{U}(0, 10)$ | **95.78** | 360 |
| | CS | $\mathcal{U}(0, 10)$ | 98.53 | 347 |
| | CS-aligned | $\mathcal{U}(0, 10)$ | 97.89 | 358 |
| 12951 | US | $\mathcal{U}(0, 10)$ | 93.38 | 364 |
| | CS | $\mathcal{U}(0, 10)$ | 93.15 | 348 |
| | CS-aligned | $\mathcal{U}(0, 10)$ | **92.91** | 354 |

Table 17: Comparison of uniform sampling and our coreset method using TSP100-3D-$\mathcal{U}(0, 10)$ as the training dataset on test data of varying sizes. We fix the sample size as 12951.

| TSP size | Method | Test distribution | Greedy | |
|---|---|---|---|---|
| | | | Length (↓) | Time (↓) |
| TSP-200 | Org | $\mathcal{U}(0, 10)$ | 125.98 | 73 |
| TSP-500 | Org | $\mathcal{U}(0, 10)$ | 210.15 | 674 |
| TSP-1000 | Org | $\mathcal{U}(0, 10)$ | 313.00 | 2818 |
| TSP-200 | US | $\mathcal{U}(0, 10)$ | 145.67 | 78 |
| | CS | $\mathcal{U}(0, 10)$ | 144.27 | 76 |
| | CS-aligned | $\mathcal{U}(0, 10)$ | **143.46** | 75 |
| TSP-500 | US | $\mathcal{U}(0, 10)$ | 302.16 | 674 |
| | CS | $\mathcal{U}(0, 10)$ | 244.63 | 676 |
| | CS-aligned | $\mathcal{U}(0, 10)$ | **236.33** | 689 |
| TSP-1000 | US | $\mathcal{U}(0, 10)$ | 535.90 | 2815 |
| | CS | $\mathcal{U}(0, 10)$ | 367.04 | 2812 |
| | CS-aligned | $\mathcal{U}(0, 10)$ | **308.71** | 2816 |

Table 18: Time statistics for different phases of MIS.

| Method | Sample size | Labeling time | Coreset Time | Training Time | Total time |
|--------|-------------|---------------|--------------|---------------|------------|
| Org | 128000 | 102600 | - | 16762 | 119362 |
| US | 3973 | 3184 | - | 1726 | 4910 |
| | 8001 | 6413 | - | 2503 | 8916 |
| | 12417 | 9953 | - | 3400 | 13353 |
| CS | 3973 | 4007 | 2263 | 3227 | 9497 |
| | 8001 | 7322 | 2424 | 5031 | 14777 |
| | 12417 | 11032 | 2511 | 7011 | 20554 |

Table 19: Comparison of uniform sampling and our coreset method with training dataset ER-[90-100] on test data from different distributions.

| Sample size | Method | Test distribution | Greedy | | Sampling | |
|-------------|--------|-------------------|--------|--------|----------|--------|
| | | | Size (↑) | Time (↓) | Size (↑) | Time (↓) |
| 128000 | Org | ER-[v90-100] | 22.34 | 64 | 23.27 | 70 |
| | | ER-[400-500] | 34.66 | 134 | 36.69 | 551 |
| | | ER-[700-800] | 37.51 | 393 | 40.06 | 1463 |
| 3973 | US | ER-[v90-100] | 19.57 | 64 | 21.34 | 319 |
| | | ER-[400-500] | 27.84 | 134 | 29.48 | 552 |
| | | ER-[700-800] | 30.30 | 391 | 32.23 | 1469 |
| | CS | ER-[v90-100] | **19.77** | 63 | **21.49** | 72 |
| | | ER-[400-500] | **27.99** | 135 | **29.85** | 558 |
| | | ER-[700-800] | **30.61** | 392 | **32.38** | 1524 |
| 8001 | US | ER-[v90-100] | **20.38** | 64 | **22.33** | 70 |
| | | ER-[400-500] | 28.81 | 135 | 30.59 | 554 |
| | | ER-[700-800] | 31.28 | 393 | 32.77 | 1471 |
| | CS | ER-[v90-100] | 20.25 | 64 | 22.27 | 73 |
| | | ER-[400-500] | **29.59** | 134 | **31.91** | 549 |
| | | ER-[700-800] | **32.23** | 393 | **33.93** | 1460 |
| 12417 | US | ER-[v90-100] | 20.80 | 63 | 22.30 | 71 |
| | | ER-[400-500] | 30.36 | 135 | 32.55 | 549 |
| | | ER-[700-800] | 32.30 | 393 | 34.25 | 1458 |
| | CS | ER-[v90-100] | **21.04** | 63 | **22.55** | 70 |
| | | ER-[400-500] | **31.00** | 135 | **33.11** | 548 |
| | | ER-[700-800] | **32.96** | 393 | **35.59** | 1462 |

Table 20: Comparison of uniform sampling and our coreset method with training dataset ER-[90-100] on test data SATLIB.

| Sample size | Method | Test distribution | Greedy | | Sampling | |
|-------------|--------|-------------------|--------|--------|----------|--------|
| | | | Size (↑) | Time (↓) | Size (↑) | Time (↓) |
| 128000 | Org | ER-[v90-100] | 22.34 | 64 | 23.27 | 70 |
| 3973 | US | SATLIB | **1015.64** | 120 | **1022.81** | 490 |
| | CS | SATLIB | 1015.31 | 122 | 1021.24 | 518 |
| 8245 | US | SATLIB | **410.61** | 296 | **413.49** | 898 |
| | CS | SATLIB | 409.87 | 299 | 413.34 | 902 |
| 12417 | US | SATLIB | 410.32 | 304 | 413.55 | 906 |
| | CS | SATLIB | **410.83** | 297 | **414.04** | 900 |

Table 21: Comparison of uniform sampling and our coreset method with training dataset CVRP100-$\mathcal{N}(0, 0.1^2)$ on test data from different distributions.

| Sample size | Method | Test distribution | RRC-budget (↓) | Length (↓) | Gap (↓) | Time (↓) |
|---|---|---|---|---|---|---|
| 128000 | Org | CVRP100 | 0 | 17.64 | 6.34% | 3 |
| | | | 50 | 16.58 | -0.02% | 45 |
| | | | 100 | 16.43 | -0.96% | 86 |
| | | | 200 | 16.30 | -1.70% | 162 |
| | | | 500 | 16.18 | -2.43% | 397 |
| 4437 | US | CVRP100 | 0 | 19.56 | 17.91% | 3 |
| | | | 50 | 17.85 | 7.64% | 45 |
| | | | 100 | 17.57 | 5.96% | 86 |
| | | | 200 | 17.30 | 4.28% | 162 |
| | | | 500 | 17.03 | 2.71% | 397 |
| | CS | CVRP100 | 0 | **19.16** | **15.51%** | 3 |
| | | | 50 | **17.65** | **6.41%** | 45 |
| | | | 100 | **17.38** | **4.78%** | 86 |
| | | | 200 | **17.15** | **3.38%** | 163 |
| | | | 500 | **16.91** | **2.01%** | 400 |
| 8082 | US | CVRP100 | 0 | 18.77 | 13.18% | 3 |
| | | | 50 | 17.35 | 4.62% | 47 |
| | | | 100 | 17.13 | 3.27% | 91 |
| | | | 200 | 16.92 | 2.01% | 172 |
| | | | 500 | 16.71 | 0.75% | 423 |
| | CS | CVRP100 | 0 | **18.49** | **11.51%** | 3 |
| | | | 50 | **17.19** | **3.66%** | 45 |
| | | | 100 | **16.98** | **2.38%** | 88 |
| | | | 200 | **16.78** | **1.19%** | 168 |
| | | | 500 | **16.60** | **0.06%** | 413 |
| 12175 | US | CVRP100 | 0 | 18.65 | 12.47% | 3 |
| | | | 50 | 17.24 | 3.95% | 47 |
| | | | 100 | 17.01 | 2.59% | 90 |
| | | | 200 | 16.82 | 1.43% | 170 |
| | | | 500 | 16.63 | 0.25% | 415 |
| | CS | CVRP100 | 0 | **18.53** | **11.72%** | 3 |
| | | | 50 | **17.18** | **3.56%** | 47 |
| | | | 100 | **16.96** | **2.23%** | 90 |
| | | | 200 | **16.77** | **1.13%** | 170 |
| | | | 500 | **16.58** | **-0.01%** | 415 |

test data for other cases. Obviously, the uniform distribution has the highest entropy and thus the highest diversity. For these Gaussian distributions, the larger the variance, the larger the diversity.

**Results of CVRP100** Tables 21 to 23 present the results on training dataset CVRP100-$\mathcal{N}(0, 0.1^2)$. The training datasets are generated by the uniform sampling and our coreset technique respectively. The results show that both our method and uniform sampling method perform better as the sample size increases. For both the test data with distribution shift and without distribution shift, our method consistently shows better performance. Table 22 further illustrates that our method can generalize to large-scale CVRP problems better. Table 23 shows the performance on the CVRPLIB dataset.

Tables 24 to 26 present the results on training dataset CVRP100-$\mathcal{U}(0, 1)$. The training datasets are generated by the uniform sampling and our coreset technique respectively. The results show that both our method and uniform sampling method perform better as the sample size increases. For both the test data with distribution shift and without distribution shift, our method consistently shows better performance. Table 25 further illustrates that our method can generalize to large-scale CVRP problems better. Table 26 shows the performance on the CVRPLIB dataset.

Table 22: Comparison of uniform sampling and our coreset method with training dataset CVRP100-$\mathcal{N}(0, 0.1^2)$ on test data of varying sizes. We fix the sample size as 12175.

| Sample size | Method | Test distribution | RRC-budget ($\downarrow$) | Length ($\downarrow$) | Gap ($\downarrow$) | Time ($\downarrow$) |
|---|---|---|---|---|---|---|
| 128000 | Org | CVRP200 | 0 | 22.70 | 12.52% | 2 |
| | | | 50 | 21.50 | 6.60% | 26 |
| | | | 100 | 21.34 | 5.77% | 49 |
| | | | 200 | 21.16 | 4.90% | 105 |
| | | | 500 | 20.96 | 3.88% | 256 |
| | | CVRP500 | 0 | 41.68 | 11.95% | 2 |
| | | | 50 | 40.23 | 8.06% | 179 |
| | | | 100 | 39.91 | 7.20% | 298 |
| | | | 200 | 39.64 | 6.48% | 604 |
| | | | 500 | 39.32 | 5.62% | 1628 |
| | | CVRP1000 | 0 | 44.62 | 20.29% | 73 |
| | | | 50 | 42.74 | 15.25% | 895 |
| | | | 100 | 42.36 | 14.22% | 1810 |
| | | | 200 | 41.81 | 12.72% | 3672 |
| | | | 500 | 41.19 | 11.04% | 9108 |
| 12175 | US | CVRP200 | 0 | 24.17 | 19.80 % | 2 |
| | | | 50 | 22.79 | 12.97% | 26 |
| | | | 100 | 22.52 | 11.61% | 50 |
| | | | 200 | 22.24 | 10.27% | 108 |
| | | | 500 | 21.89 | 8.52% | 263 |
| | CS | CVRP200 | 0 | **24.00** | **18.97%** | 2 |
| | | | 50 | **22.60** | **12.03%** | 25 |
| | | | 100 | **22.36** | **10.83%** | 48 |
| | | | 200 | **22.13** | **9.71%** | 105 |
| | | | 500 | **21.75** | **7.84%** | 256 |
| 12175 | US | CVRP500 | 0 | 45.23 | 21.50% | 2 |
| | | | 50 | 43.40 | 16.58% | 173 |
| | | | 100 | 42.84 | 15.06% | 288 |
| | | | 200 | 42.39 | 13.85% | 584 |
| | | | 500 | 41.82 | 12.32% | 1579 |
| | CS | CVRP500 | 0 | **45.10** | **21.15%** | 2 |
| | | | 50 | **43.19** | **16.00%** | 173 |
| | | | 100 | **42.73** | **14.78%** | 287 |
| | | | 200 | **42.28** | **13.56%** | 583 |
| | | | 500 | **41.64** | **11.85%** | 1576 |

Continued on next page

Table 22: Comparison of uniform sampling and our coreset method with training dataset CVRP100-$\mathcal{N}(0, 0.1^2)$ on test data of varying sizes. We fix the sample size as 12175.

| Sample size | Method | Test distribution | RRC-budget ($\downarrow$) | Length ($\downarrow$) | Gap ($\downarrow$) | Time ($\downarrow$) |
|---|---|---|---|---|---|---|
| 12175 | US | CVRP1000 | 0 | 53.01 | 42.91% | 74 |
| | | | 50 | 49.42 | 33.23% | 897 |
| | | | 100 | 48.81 | 31.60% | 1819 |
| | | | 200 | 47.95 | 29.28% | 3708 |
| | | | 500 | 46.91 | 26.48% | 9180 |
| | CS | CVRP1000 | 0 | 51.88 | 39.88% | 73 |
| | | | 50 | 48.24 | 30.06% | 892 |
| | | | 100 | 47.71 | 28.64% | 1805 |
| | | | 200 | 47.00 | 26.71% | 3672 |
| | | | 500 | 46.12 | 24.34% | 9108 |

Table 23: Comparison of uniform sampling and our coreset method with training dataset CVRP100-$\mathcal{N}(0, 0.1^2)$ on test data CVRPLib.

| Sample size | Method | Test distribution | RRC-budget ($\downarrow$) | Length ($\downarrow$) | Gap ($\downarrow$) | Time ($\downarrow$) |
|---|---|---|---|---|---|---|
| 128000 | Org | CVRPLib | 0 | 913.17 | 16.48% | 1 |
| | | | 50 | 787.20 | 0.41% | 4 |
| | | | 100 | 787.20 | 0.41% | 7 |
| | | | 200 | 787.20 | 0.41% | 12 |
| | | | 500 | 787.20 | 0.41% | 29 |
| 4437 | US | CVRPLib | 0 | **927.95** | **18.36%** | 1 |
| | | | 50 | 868.66 | 10.80% | 4 |
| | | | 100 | 864.51 | 10.27% | 7 |
| | | | 200 | 844.59 | 7.73% | 14 |
| | | | 500 | 844.59 | 7.73% | 33 |
| | CS | CVRPLib | 0 | 950.06 | 21.18% | 1 |
| | | | 50 | **841.93** | **7.39%** | 4 |
| | | | 100 | **837.50** | **6.82%** | 7 |
| | | | 200 | **837.50** | **6.82%** | 14 |
| | | | 500 | **834.28** | **6.41%** | 32 |
| 8082 | US | CVRPLib | 0 | 971.82 | 23.96% | 1 |
| | | | 50 | 867.72 | 10.68% | 4 |
| | | | 100 | **796.05** | **1.54%** | 7 |
| | | | 200 | **796.05** | **1.54%** | 13 |
| | | | 500 | **787.81** | **0.49%** | 30 |
| | CS | CVRPLib | 0 | **917.07** | **16.97%** | 1 |
| | | | 50 | **829.60** | **5.82%** | 4 |
| | | | 100 | 829.60 | 5.82% | 7 |
| | | | 200 | 823.85 | 5.08% | 13 |
| | | | 500 | 791.81 | 1.00% | 30 |
| 12175 | US | CVRPLib | 0 | 1016.27 | 29.63 % | 1 |
| | | | 50 | 865.31 | 10.37% | 4 |
| | | | 100 | 865.31 | 10.37% | 7 |
| | | | 200 | 858.44 | 9.49% | 13 |
| | | | 500 | 858.44 | 9.49% | 30 |
| | CS | CVRPLib | 0 | **917.09** | **16.98%** | 1 |
| | | | 50 | **811.18** | **3.47%** | 4 |
| | | | 100 | **810.27** | **3.35%** | 7 |
| | | | 200 | **806.12** | **2.82%** | 13 |
| | | | 500 | **806.12** | **2.82%** | 30 |

Table 24: Comparison of uniform sampling and our coreset method with training dataset CVRP100-$\mathcal{U}(0,1)$ on test data from different distributions.

| Sample size | Method | Test distribution | RRC-budget ($\downarrow$) | Length ($\downarrow$) | Gap ($\downarrow$) | Time ($\downarrow$) |
|---|---|---|---|---|---|---|
| 128000 | Org | CVRP100 | 0 | 17.35 | 4.64% | 3 |
| | | | 50 | 16.36 | -1.38% | 47 |
| | | | 100 | 16.22 | -2.18% | 90 |
| | | | 200 | 16.13 | -2.75% | 170 |
| | | | 500 | 16.02 | -3.40% | 418 |
| 4697 | US | CVRP100 | 0 | 18.48 | 11.43 % | 3 |
| | | | 50 | 17.15 | 3.39% | 44 |
| | | | 100 | 16.93 | 2.08% | 85 |
| | | | 200 | 16.74 | 0.93% | 161 |
| | | | 500 | 16.57 | -0.10% | 396 |
| | CS | CVRP100 | 0 | **18.44** | **11.20%** | 3 |
| | | | 50 | **17.10** | **3.10%** | 44 |
| | | | 100 | **16.89** | **1.85%** | 85 |
| | | | 200 | **16.70** | **0.69%** | 161 |
| | | | 500 | **16.53** | **-0.34%** | |
| 7694 | US | CVRP100 | 0 | 18.22 | 9.83% | 3 |
| | | | 50 | **16.91** | **1.96%** | 44 |
| | | | 100 | **16.71** | **0.78%** | 83 |
| | | | 200 | **16.55** | **-0.21%** | 158 |
| | | | 500 | **16.39** | **-1.17%** | 389 |
| | CS | CVRP100 | 0 | **18.16** | **9.52%** | 3 |
| | | | 50 | 16.91 | 1.97% | 45 |
| | | | 100 | 16.72 | 0.82% | 86 |
| | | | 200 | 16.56 | -0.18% | 162 |
| | | | 500 | 16.40 | -1.10% | 400 |
| 12033 | US | CVRP100 | 0 | 18.03 | 8.70% | 3 |
| | | | 50 | 16.82 | 1.41% | 45 |
| | | | 100 | 16.64 | 0.34% | 86 |
| | | | 200 | 16.50 | -0.54% | 162 |
| | | | 500 | 16.35 | -1.43% | 395 |
| | CS | CVRP100 | 0 | **18.01** | **8.59%** | 3 |
| | | | 50 | **16.81** | **1.36%** | 44 |
| | | | 100 | **16.62** | **0.21%** | 84 |
| | | | 200 | **16.48** | **-0.64%** | 160 |
| | | | 500 | **16.34** | **-1.55%** | 39 |

Table 25: Comparison of uniform sampling and our coreset method with training dataset CVRP100-$\mathcal{U}(0,1)$ on test data of varying sizes. We fix the sample size as 12033.

| Sample size | Method | Test distribution | RRC-budget ($\downarrow$) | Length ($\downarrow$) | Gap ($\downarrow$) | Time ($\downarrow$) |
|---|---|---|---|---|---|---|
| 128000 | Org | CVRP200 | 0 | 22.41 | 11.08% | 2 |
| | | | 50 | 21.20 | 5.08% | 36 |
| | | | 100 | 21.05 | 4.36% | 68 |
| | | | 200 | 20.93 | 3.76% | 145 |
| | | | 500 | 20.75 | 2.88% | 353 |
| | | CVRP500 | 0 | 41.01 | 10.16% | 1 |
| | | | 50 | 39.69 | 6.61% | 190 |
| | | | 100 | 39.42 | 5.87% | 319 |
| | | | 200 | 39.16 | 5.20% | 645 |
| | | | 500 | 38.91 | 4.51% | 1730 |
| | | CVRP1000 | 0 | 43.09 | 16.19% | 73 |

Continued on next page

Table 25: Comparison of uniform sampling and our coreset method with training dataset CVRP100-$\mathcal{U}(0,1)$ on test data of varying sizes. We fix the sample size as 12033.

| Sample size | Method | Test distribution | RRC-budget (↓) | Length (↓) | Gap (↓) | Time (↓) |
|---|---|---|---|---|---|---|
| | | | 50 | 37.09 | 11.85% | 904 |
| | | | 100 | 41.17 | 11.00% | 1828 |
| | | | 200 | 40.76 | 9.90% | 3708 |
| | | | 500 | 40.29 | 8.63% | 9216 |
| 12033 | US | CVRP200 | 0 | 23.29 | 15.45% | 2 |
| | | | 50 | 21.98 | 8.94% | 28 |
| | | | 100 | **21.75** | **7.82%** | 54 |
| | | | 200 | **21.57** | **6.91%** | 115 |
| | | | 500 | 21.33 | 5.72% | 281 |
| | CS | CVRP200 | 0 | **20.17** | **14.97%** | 2 |
| | | | 50 | **21.93** | **8.70%** | 29 |
| | | | 100 | 21.75 | 7.83% | 54 |
| | | | 200 | 21.59 | 7.04% | 116 |
| | | | 500 | **21.30** | **5.50%** | 282 |
| 12033 | US | CVRP500 | 0 | 37.23 | 16.76% | 2 |
| | | | 50 | 41.77 | 12.19% | 177 |
| | | | 100 | 41.38 | 11.15% | 296 |
| | | | 200 | 41.02 | 10.18% | 598 |
| | | | 500 | 40.51 | 8.82% | 1620 |
| | CS | CVRP500 | 0 | **43.19** | **16.02%** | 2 |
| | | | 50 | **41.66** | **11.89%** | 176 |
| | | | 100 | **41.25** | **10.81%** | 295 |
| | | | 200 | **40.88** | **9.80%** | 597 |
| | | | 500 | **40.40** | **8.52%** | 1609 |
| 12033 | US | CVRP1000 | 0 | **48.20** | **29.96%** | 73 |
| | | | 50 | 46.12 | 24.33% | 907 |
| | | | 100 | 45.67 | 23.14% | 1847 |
| | | | 200 | 45.04 | 21.43% | 3816 |
| | | | 500 | 44.29 | 19.42% | 9324 |
| | CS | CVRP1000 | 0 | 48.24 | 30.05% | 73 |
| | | | 50 | **45.89** | **23.73%** | 899 |
| | | | 100 | **45.50** | **22.66%** | 1820 |
| | | | 200 | **44.87** | **21.02%** | 3672 |
| | | | 500 | **44.07** | **18.82%** | 9180 |

Table 26: Comparison of uniform sampling and our coreset method with training dataset CVRP100-$\mathcal{U}(0, 1)$ on test data CVRPLib.

| Sample size | Method | Test distribution | RRC-budget ($\downarrow$) | Length ($\downarrow$) | Gap ($\downarrow$) | Time ($\downarrow$) |
|---|---|---|---|---|---|---|
| 128000 | Org | CVRPLib | 0 | 856.40 | 9.24% | 1 |
| | | | 50 | 797.45 | 1.72% | 6 |
| | | | 100 | 797.45 | 1.72% | 10 |
| | | | 200 | 797.45 | 1.72% | 20 |
| | | | 500 | 797.45 | 1.72% | 47 |
| 4697 | US | CVRPLib | 0 | 902.29 | 15.09% | 1 |
| | | | 50 | 846.35 | 7.95% | 5 |
| | | | 100 | 838.10 | 6.90% | 7 |
| | | | 200 | 838.10 | 6.90% | 14 |
| | | | 500 | 838.10 | 6.90% | 32 |
| | CS | CVRPLib | 0 | 1020.05 | 30.10% | 1 |
| | | | 50 | 809.93 | 3.30% | 4 |
| | | | 100 | 797.45 | 1.72% | 7 |
| | | | 200 | 797.45 | 1.72% | 14 |
| | | | 500 | 797.45 | 1.72% | 32 |
| 8082 | US | CVRPLib | 0 | 1096.32 | 39.84% | 1 |
| | | | 50 | 846.93 | 8.03% | 4 |
| | | | 100 | 846.93 | 8.03% | 7 |
| | | | 200 | 846.93 | 8.03% | 13 |
| | | | 500 | 846.93 | 8.03% | 32 |
| | CS | CVRPLib | 0 | 866.09 | 10.47% | 1 |
| | | | 50 | 789.79 | 0.74% | 4 |
| | | | 100 | 789.79 | 0.74% | 7 |
| | | | 200 | 789.79 | 0.74% | 13 |
| | | | 500 | 789.79 | 0.74% | 31 |
| 12033 | US | CVRPLib | 0 | 864.85 | 10.31% | 1 |
| | | | 50 | 830.51 | 5.93% | 4 |
| | | | 100 | 797.45 | 1.72% | 7 |
| | | | 200 | 797.45 | 1.72% | 14 |
| | | | 500 | 797.45 | 1.72% | 32 |
| | CS | CVRPLib | 0 | **965.89** | **23.20%** | 1 |
| | | | 50 | **816.14** | **4.10%** | 4 |
| | | | 100 | **809.71** | **3.28%** | 7 |
| | | | 200 | **799.16** | **1.93%** | 13 |
| | | | 500 | **787.20** | **0.41%** | 32 |

Table 27: Comparison of uniform sampling and our coreset method with different graph embedding techniques on test data from different distributions. CS-spring is the embedding technique based on force-directed representation; CS-spectral is the spectral embedding technique; CS-MDS is the embedding technique based on multidimensional scaling.

| Sample size | Method | Test distribution | Size ($\uparrow$) | Time ($\downarrow$) |
|---|---|---|---|---|
| 4010 | US | ER-[400-500] | 27.40 | 133 |
| | | ER-[700-800] | 30.36 | 392 |
| | | ER-[1400-1500] | 34.05 | 1361 |
| 3973 | CS-spring | ER-[400-500] | 28.46 | 135 |
| | | ER-[700-800] | 30.89 | 389 |
| | | ER-[1400-1500] | 34.25 | 1361 |
| 8001 | CS-spectral | ER-[400-500] | 27.68 | 132 |
| | | ER-[700-800] | 30.43 | 391 |
| | | ER-[1400-1500] | 34.14 | 1362 |
| 12417 | CS-MDS | ER-[400-500] | 28.43 | 132 |
| | | ER-[700-800] | 31.10 | 389 |
| | | ER-[1400-1500] | 34.52 | 1361 |

