# OpenReview forum: "Efficient and Robust Neural Combinatorial Optimization via Wasserstein-Based Coresets"
_ICLR.cc/2025/Conference — ICLR 2025 Poster_

### Official Review · Reviewer_wmHE · 2024-10-29

**Soundness:** 3
**Presentation:** 3
**Contribution:** 2
**Rating:** 6
**Confidence:** 3

**Summary:**

The authors represent CO instances as probability measures and utilize a variant of Wasserstein distance that accounts for rigid transformations, allowing the model to better handle similar instances.

They construct a small, representative subset of the original data using hierarchical clustering. This process is optimized using a "merge-and-reduce" framework, enabling parallel computing and faster training times.

The proposed framework replaces large datasets with the coreset for training, reducing computational and storage needs. For inference, the framework aligns test data along a hierarchical tree, which further accelerates the process.

Through experiments on tasks such as the Traveling Salesperson Problem (TSP), the authors demonstrate that the coreset-based framework achieves robust performance and efficiency gains, especially under distribution shifts.

**Strengths:**

Originality

This paper introduces a novel approach to neural combinatorial optimization (NCO) by leveraging Wasserstein-based coresets to address resource and robustness challenges in training models for combinatorial optimization problems. The originality stems from: 1. Modeling combinatorial optimization instances as probability measures and applying the Wasserstein distance under rigid transformations (RWD) is innovative. This framework addresses the challenges of data redundancy and distribution shifts, providing a fresh perspective in the NCO space. 2. While coresets are common in clustering and other machine learning tasks, adapting this technique for combinatorial optimization with merge-and-reduce frameworks for accelerated parallel computing is both creative and practical.

Quality

The paper is well-executed, with rigorous theoretical foundations and a thoughtful experimental setup: 1. The authors provide a solid mathematical foundation for the proposed coreset construction, including formal definitions, theoretical guarantees, and proof sketches that validate the coreset’s ability to represent large datasets with minimal error. 2. The experiments cover various TSP instances under both uniform sampling and coreset techniques, testing robustness to distribution shifts and different dimensions (2D and 3D) with insightful comparisons across sample sizes and distributions. This comprehensive approach substantiates the paper's claims.

Clarity

The paper is well-structured and communicates its ideas effectively: 1. The paper follows a clear progression from defining the problem and introducing the method to presenting experimental results. Definitions, such as the RWD and coreset construction, are introduced at appropriate points, aiding understanding. 2. Algorithmic steps are outlined in detail, making it easier to understand implementation specifics, especially for constructing coresets.

Significance

This work has substantial potential significance for both research and practical applications: 1. By enabling training on compact coresets, the paper addresses one of the most significant limitations in neural combinatorial optimization—computational inefficiency with large datasets. This is a notable advancement that could make NCO more feasible in real-world, large-scale applications like logistics, robotics, and manufacturing. 2. The paper’s robustness improvements are important for applications where data distributions vary, making it applicable across domains where data acquisition is not consistently distributed or is prone to changes over time.

**Weaknesses:**

1. The coreset construction approach offers impressive results in reducing data size while preserving accuracy. A deeper discussion on the trade-offs between dataset compression and accuracy loss would be beneficial. Although error bounds are mentioned, it would strengthen the paper if experiments explicitly analyzed performance degradation as coreset size is reduced.

2. It’s unclear how the coreset method performs across diverse types of CO instances (e.g., sparse vs. dense graphs or varying clustering characteristics). Adding experiments with different dataset properties could clarify the method’s robustness and practical applicability.

3.  Aligning instances under rigid transformations for each test instance could be computationally expensive, especially for larger datasets or higher-dimensional instances (e.g., TSP-3D and beyond). Although a heuristic for alignment is provided, it would be helpful to quantify the computational cost and compare it to baseline methods.

4. The paper does not include a performance comparison between exact and heuristic alignment. Such an evaluation would help clarify the practical efficiency and accuracy trade-offs of the heuristic.

5. The experimental validation focuses heavily on TSP or MIS dataset, which, while effective, may limit the generalizability claims. While these are well-known CO problems, the approach could benefit from validation on other types of CO tasks with varying structures, such as graph partitioning problems. Since these problems exhibit different graph characteristics, they would better demonstrate the adaptability of the coreset method.

6. Current experiments primarily focus on tour length and runtime as evaluation metrics. Additional measures, such as robustness to noise or perturbations in the data, could further demonstrate the coreset’s value in handling real-world data variability.

7. The merge-and-reduce framework is crucial for scaling the coreset method, but the paper provides limited guidance on its implementation and scalability implications: Although the paper briefly mentions time complexity, a more detailed breakdown of the merge-and-reduce framework’s complexity across layers and for different dataset sizes would be helpful.

8. There is limited discussion on how the framework’s parallelization could be optimized or applied to larger datasets. This discussion would be especially useful for practitioners seeking to apply the method in large-scale real-world settings.

9. While the paper claims improved robustness to distribution shifts through the use of coresets, this aspect is not rigorously analyzed or compared. The paper could benefit from quantifying the robustness improvements by comparing the method’s performance across significantly different distributions and measuring accuracy decay.

**Questions:**

see weakness

---

> ### Author Response · Authors · 2024-11-20
> **Rebuttal by authors**
>
> > **[Q1] The coreset construction approach offers impressive results in reducing data size while preserving accuracy. A deeper discussion on the trade-offs between dataset compression and accuracy loss would be beneficial. Although error bounds are mentioned, it would strengthen the paper if experiments explicitly analyzed performance degradation as coreset size is reduced.**
>
>
> Thank you for your insightful advice. We will provide an explicit analysis of performance degradation as the coreset size is reduced.
>
> In our experiments, we have added the analyses across various sample sizes, as reflected in Tables 1, 3, 5, 6, 8, 9 and 10. For convenience, I take Table 1 as an example.
>
> In Table 1, we report the results of different models trained by using different training sample sizes (e.g., 4003, 8245, 12951). These results demonstrate that as the sample size increases, the performance improves for both our coreset methods (CS) and the uniform sampling (US) baseline. Furthermore, our methods consistently outperform the US methods across all sample sizes.
>
> Notably, as the sample size decreases, the advantage of our methods compared to uniform sampling becomes increasingly evident.
>
> ---
>
> ---
>
> **Table 1:** Comparison of uniform sampling and our coreset method using TSP100-2D-𝒩(0, 1) as the training dataset on test data TSP100-2D from different distributions.
>
> | Sample size | Method       | Test distribution | Greedy Length (↓) | Greedy Time (↓) | Greedy+2-opt Length (↓) | Greedy+2-opt Time (↓) |
> |-------------|--------------|-------------------|--------------------|-----------------|--------------------------|-----------------------|
> | 4003        | US           | 𝒩(0, 1)          | 22.34             | 378             | 18.92                   | 387                   |
> |             |              | 𝒩(0, 4)          | 101.95            | 379             | 69.28                   | 395                   |
> |             |              | 𝒰(0, 10)         | 119.78            | 380             | 82.59                   | 395                   |
> |             | CS           | 𝒩(0, 1)          | **22.21**             | 372             | **18.87**                   | 379                   |
> |             |              | 𝒩(0, 4)          | **80.63**             | 372             | **67.92**                   | 379                   |
> |             |              | 𝒰(0, 10)         | **94.73**             | 373             | **80.64**                   | 377                   |
> |-------------|--------------|-------------------|--------------------|-----------------|--------------------------|-----------------------|
> | 8245        | US           | 𝒩(0, 1)          | 22.12             | 377             | 18.87                   | 388                   |
> |             |              | 𝒩(0, 4)          | 83.17             | 377             | 68.13                   | 378                   |
> |             |              | 𝒰(0, 10)         | 97.31             | 377             | 80.80                   | 387                   |
> |             | CS           | 𝒩(0, 1)          | **21.79**             | 366             | **18.84**                   | 383                   |
> |             |              | 𝒩(0, 4)          | **78.72**             | 372             |**67.79**                   | 378                   |
> |             |              | 𝒰(0, 10)         | **92.99**             | 374             | **80.35**                   | 377                   |
> |-------------|--------------|-------------------|--------------------|-----------------|--------------------------|-----------------------|
> | 12951       | US           | 𝒩(0, 1)          | 21.99             | 390             | 18.87                   | 377                   |
> |             |              | 𝒩(0, 4)          | 80.78             | 384             | 67.94                   | 379                   |
> |             |              | 𝒰(0, 10)         | 95.01             | 379             | 80.60                   | 379                   |
> |             | CS           | 𝒩(0, 1)          |**21.57**             | 372             | **18.81**                   | 382                   |
> |             |              | 𝒩(0, 4)          | **77.80**             | 369             | **67.58**                   | 379                   |
> |             |              | 𝒰(0, 10)         | **92.01**             | 378             | **80.23**                   | 375                   |
> ｜
>
> We will add explicit discussions in the revised manuscript to highlight these observations. Thank you once again for pointing out this opportunity to enhance our presentation.

---

> ### Author Response · Authors · 2024-11-20
> **Rebuttal by authors**
>
> > **[Q2] It’s unclear how the coreset method performs across diverse types of CO instances (e.g., sparse vs. dense graphs or varying clustering characteristics). Adding experiments with different dataset properties could clarify the method’s robustness and practical applicability.**
>
>
> Thank you for your suggestion. We are currently working on experiments related to graph partitioning, and if time permits, we will include these in the revised version later.
>
> Meanwhile, in the past month, we have added experiments on the CVRP dataset. We take Table 22 as an example to show the performance on CVRP dataset. (Additional results are available in Tables 23, 24, and 25 in the Appendix.) As shown in Table 21, our method consistently performs better.
>
> ---
>
> ---
> **Table 22:** Comparison of uniform sampling and our coreset method with training dataset CVRP100-𝒩(0, 0.1) on test data from different distributions.
>
> | Sample size | Method | Test distribution | Length (↓) | Gap (↓) | Time (↓) |
> |-------------|--------|-------------------|----------------|---------|----------|
> | 128000      | Org    | CVRP100           | 17.64          | 6.34%   | 3        |
> |             |        |                   | 16.58          | -0.02%  | 45       |
> |             |        |                   | 16.43          | -0.96%  | 86       |
> |             |        |                   | 16.30          | -1.70%  | 162      |
> |             |        |                   | 16.18          | -2.43%  | 397      |
> |-------------|--------|-------------------|----------------|---------|----------|
> | 4437        | US     | CVRP100           | 19.56          | 17.91%  | 3        |
> |             |        |                   | 17.35          | 7.64%   | 45       |
> |             |        |                   | 17.57          | 5.96%   | 86       |
> |             |        |                   | 17.30          | 4.28%   | 162      |
> |             |        |                   | 17.03          | 2.71%   | 397      |
> |-------------|--------|-------------------|----------------|---------|----------|
> | 4437        | CS | CVRP100           | **19.16**      | 15.51%  | 3    |
> |             |        |                   | **17.65**      | 6.41%   | 45   |
> |             |        |                   | **17.38**      | 4.78%   | 86   |
> |             |        |                   | **17.15**      | 3.38%   | 163  |
> |             |        |                   | **17.03**      | 2.01%   | 400  |
> |-------------|--------|-------------------|----------------|---------|----------|
> | 8082        | US     | CVRP100           | 18.77          | 13.18%  | 3        |
> |             |        |                   | 17.35          | 4.62%   | 47       |
> |             |        |                   | 17.13          | 3.27%   | 91       |
> |             |        |                   | 16.92          | 2.01%   | 172      |
> |             |        |                   | 16.71          | 0.75%   | 423      |
> |-------------|--------|-------------------|----------------|---------|----------|
> | 8082        | CS | CVRP100           | **18.49**      | 11.51%  | 3    |
> |             |        |                   | **17.19**      | 3.66%   | 45   |
> |             |        |                   | **16.98**      | 2.38%   | 88   |
> |             |        |                   | **16.78**      | 1.19%   | 168  |
> |             |        |                   | **16.60**      | 0.06%   | 413  |
> |-------------|--------|-------------------|----------------|---------|----------|
> | 12175       | US     | CVRP100           | 18.65          | 12.47%  | 3        |
> |             |        |                   | 17.24          | 3.95%   | 47       |
> |             |        |                   | 17.01          | 2.59%   | 90       |
> |             |        |                   | 16.82          | 1.43%   | 170      |
> |             |        |                   | 16.63          | 0.25%   | 415      |
> |-------------|--------|-------------------|----------------|---------|----------|
> | 12175       | CS | CVRP100           | **18.53**      | 11.72%  | 3   |
> |             |        |                   | **17.13**      | 3.56%   | 47   |
> |             |        |                   | **16.96**      | 2.23%   | 90   |
> |             |        |                   | **16.77**      | 1.13%   | 170  |
> |             |        |                   | **16.58**      | -0.01%  | 415  |
> |-------------|--------|-------------------|----------------|---------|----------|

---

> > ### Author Response · Authors · 2024-11-21
> > **Rebuttal by authors**
> >
> > > **[Q3] Aligning instances under rigid transformations for each test instance could be computationally expensive, especially for larger datasets or higher-dimensional instances (e.g., TSP-3D and beyond). Although a heuristic for alignment is provided, it would be helpful to quantify the computational cost and compare it to baseline methods.**
> >
> >
> > Thank you for your valuable suggestions! We take TSP-3D as an example to show the performance of our heuristic alignment method in Table 10. It shows that the computational cost of our heuristic alignment method is acceptable from the last column in Table 10. For cases involving very high dimensions, alignment can be optionally omitted. Even without alignment, our coreset method works well. Our aim was to provide alignment as a flexible choice to enhance efficiency rather than a mandatory step.
> >
> > ---
> >
> > ---
> >
> > **Table 10:** Comparison of uniform sampling and our coreset method using TSP100-3D-𝒩(0, 1) as the training dataset on test data TSP100-3D from different distributions.
> >
> > | Sample size | Method       | Test distribution | Greedy Length (↓) | Time (↓) | Alignment time (↓) |
> > |-------------|--------------|-------------------|--------------------|----------|---------------------|
> > | 4103        | US           | 𝒩(0, 1)          | 24.92             | 480      | -                   |
> > |             |              | 𝒩(0, 4)          | 96.60         | 483      | -                   |
> > |             |              | 𝒰(0, 10)         | 119.78        | 481      | -                   |
> > |-------------|--------------|-------------------|--------------------|----------|---------------------|
> > |             | CS           | 𝒩(0, 1)          | 24.89             | 364      | -                   |
> > |             |              | 𝒩(0, 4)          | 106.35            | 360      | -                   |
> > |             |              | 𝒰(0, 10)         | 111.63            | 353      | -                   |
> > |-------------|--------------|-------------------|--------------------|----------|-----------------|
> > |             | CS-aligned   | 𝒩(0, 1)          | **23.36**         | 479      | 2                   |
> > |             |              | 𝒩(0, 4)          | **91.94**             | 479      | 7                   |
> > |             |              | 𝒰(0, 10)         | **108.81**            | 483      | 11                  |
> > |-------------|--------------|-------------------|--------------------|----------|-----------------|
> > | 7960        | US           | 𝒩(0, 1)          | 23.62             | 477      | -                   |
> > |             |              | 𝒩(0, 4)          | 92.92             | 484      | -                   |
> > |             |              | 𝒰(0, 10)         | 115.62            | 481      | -                   |
> > |-------------|--------------|-------------------|--------------------|----------|-----------------|
> > |             | CS           | 𝒩(0, 1)          | 23.41             | 362      | -                   |
> > |             |              | 𝒩(0, 4)          | 86.20             | 362      | -                   |
> > |             |              | 𝒰(0, 10)         | 101.25            | 359      | -                   |
> > |-------------|--------------|-------------------|--------------------|----------|-----------------|
> > |             | CS-aligned   | 𝒩(0, 1)          | **22.83**         | 476      | 12                  |
> > |             |              | 𝒩(0, 4)          | **85.71**             | 483      | 16                  |
> > |             |              | 𝒰(0, 10)         | **99.10**             | 481      | 19                  |
> > |-------------|--------------|-------------------|--------------------|----------|----------------|
> > | 12058       | US           | 𝒩(0, 1)          | **22.10**             | 360      | -                   |
> > |             |              | 𝒩(0, 4)          | 84.28             | 368      | -                   |
> > |             |              | 𝒰(0, 10)         | 100.40            | 367      | -                   |
> > |-------------|--------------|-------------------|--------------------|----------|-----------------|
> > |             | CS           | 𝒩(0, 1)          | **22.10**             | 371      | -                   |
> > |             |              | 𝒩(0, 4)          | 80.47             | 361      | -                   |
> > |             |              | 𝒰(0, 10)         | 95.25             | 362      | -                   |
> > |-------------|--------------|-------------------|--------------------|----------|----------------|
> > |             | CS-aligned   | 𝒩(0, 1)          | 22.30         | 396      | 21                  |
> > |             |              | 𝒩(0, 4)          | **79.96**             | 388      | 23                  |
> > |             |              | 𝒰(0, 10)         | **94.13**             | 366      | 26                  |
> >
> >
> >
> > Thank you again for raising this point, and we will ensure that this is further clarified in the revised manuscript.

---

> > > ### Author Response · Authors · 2024-11-21
> > >
> > > > **[Q4] The paper does not include a performance comparison between exact and heuristic alignment. Such an evaluation would help clarify the practical efficiency and accuracy trade-offs of the heuristic.**
> > >
> > >
> > > Thank you for your valuable suggestions. We take TSP-3D as an example to compare the performance of the random heuristic alignment method (CS-rand-aligned) with our proposed alignment method（CS-aligned).
> > >
> > > As shown in Table 12, the results demonstrate that the CS-rand-aligned method performs similarly to the unaligned approach (CS), providing little improvement. In contrast, our alignment method significantly enhances performance, confirming its practical effectiveness.
> > >
> > > ---
> > >
> > > ----
> > >
> > > **Table 12:** Comparison of rand alignment method (CS-rand-aligned) and our alignment method (CS-aligned) with training dataset TSP100-3D-𝒩(0, 1) on test data of varying sizes. We fix the sample size as 12058.
> > >
> > > | TSP size | Method          | Test distribution | Greedy Length (↓) | Time (↓) | Alignment time (↓) |
> > > |----------|-----------------|-------------------|--------------------|----------|---------------------|
> > > | TSP-100  | US              | 𝒰(0, 10)         | 100.41            | 37       | -                   |
> > > |          | CS              | 𝒰(0, 10)         | 95.27             | 36       | -                   |
> > > |          | CS-aligned      | 𝒰(0, 10)         | **94.13**         | 37       | 3                   |
> > > |          | CS-rand-aligned | 𝒰(0, 10)         | 95.62             | 37       | 11                  |
> > > |----------|-----------------|-------------------|--------------------|----------|---------------------|
> > > | TSP-200  | US              | 𝒰(0, 10)         | 153.90            | 77       | -                   |
> > > |          | CS              | 𝒰(0, 10)         | 142.84            | 77       | -                   |
> > > |          | CS-aligned      | 𝒰(0, 10)         | **138.27**        | 76       | 12                  |
> > > |          | CS-rand-aligned | 𝒰(0, 10)         | 144.84            | 75       | 17                  |
> > > |----------|-----------------|-------------------|--------------------|----------|---------------------|
> > > | TSP-500  | US              | 𝒰(0, 10)         | 318.39            | 680      | -                   |
> > > |          | CS              | 𝒰(0, 10)         | 268.75            | 680      | -                   |
> > > |          | CS-aligned      | 𝒰(0, 10)         | **245.01**        | 681      | 30                  |
> > > |          | CS-rand-aligned | 𝒰(0, 10)         | 255.92            | 674      | 23                  |
> > > |----------|-----------------|-------------------|--------------------|----------|---------------------|
> > > | TSP-1000 | US              | 𝒰(0, 10)         | 550.11            | 2819     | -                   |
> > > |          | CS              | 𝒰(0, 10)         | 441.82            | 2826     | -                   |
> > > |          | CS-aligned      | 𝒰(0, 10)         | **383.92**        | 2818     | 980                 |
> > > |          | CS-rand-aligned | 𝒰(0, 10)         | 429.90            | 2817     | 493                 |
> > >
> > >
> > > We appreciate your suggestion again, which allowed us to strengthen the evaluation in our paper.

---

> ### Author Response · Authors · 2024-11-21
> **Rebuttal by authors**
>
> > **[Q6] Current experiments primarily focus on tour length and runtime as evaluation metrics. Additional measures, such as robustness to noise or perturbations in the data, could further demonstrate the coreset’s value in handling real-world data variability.**
>
>
>
> Thank you for your valuable suggestions. Indeed, the robustness to noise and perturbations is an important area for future research, and we appreciate your guidance on this matter.
>
> In the current work, we focus on the robustness to data distribution shifts and the generalization to larger problem sizes. For instance, in the case of the Traveling Salesman Problem (TSP), generalization to larger problem size refers to the ability of our method, trained on the TSP100 dataset, to perform well on larger test instances such as TSP200, TSP500, and TSP1000. As for robustness to distributional shifts, please refer to our response to the last question **[Q9]**.
>
> We use Table 2 as an example to illustrate the generalization ability of our approach. As shown, our method demonstrates better generalization to larger problem sizes.
>
> ---
>
> ---
>
> **Table 2:** Comparison of uniform sampling and our coreset method using TSP100-2D-𝒩(0, 1) as the training dataset on test data of varying sizes. We fix the sample size as 12951.
>
> | TSP size | Method      | Test distribution | Greedy Length (↓) | Time (↓) | Greedy+2-opt Length (↓) | Time (↓) |
> |----------|-------------|-------------------|--------------------|----------|--------------------------|----------|
> | TSP200   | US          | 𝒩(0, 1)          | 33.69             | 109      | 27.14                   | 112      |
> |          |             | 𝒩(0, 4)          | 125.99            | 108      | 96.70                   | 112      |
> |          |             | 𝒰(0, 10)         | 145.41            | 109      | 113.39                  | 112      |
> |          |-------------|-------------------|--------------------|----------|--------------------------|----------|
> |          | CS          | 𝒩(0, 1)          | **30.75**         | 107      | **26.69**               | 110      |
> |          |             | 𝒩(0, 4)          | **110.48**        | 109      | **94.84**               | 111      |
> |          |             | 𝒰(0, 10)         | **129.77**        | 107      | **111.47**              | 109      |
> |----------|-------------|-------------------|--------------------|----------|--------------------------|----------|
> | TSP500   | US          | 𝒩(0, 1)          | 59.81             | 1012     | 43.41                   | 1020     |
> |          |             | 𝒩(0, 4)          | 237.72            | 1012     | 154.28                  | 1022     |
> |          |             | 𝒰(0, 10)         | 263.66            | 1015     | 180.75                  | 1022     |
> |          |-------------|-------------------|--------------------|----------|--------------------------|----------|
> |       | CS          | 𝒩(0, 1)          | **49.11**         | 1010     | **42.25**               | 1016     |
> |          |             | 𝒩(0, 4)          | **178.56**        | 1010     | **149.50**              | 1016     |
> |          |             | 𝒰(0, 10)         | **208.36**        | 1011     | **174.93**              | 1016     |
> |----------|-------------|-------------------|--------------------|----------|--------------------------|----------|
> | TSP1000  | US          | 𝒩(0, 1)          | 94.71             | 2823     | 61.72                   | 2848     |
> |          |             | 𝒩(0, 4)          | 382.77            | 4224     | 219.16                  | 2847     |
> |          |             | 𝒰(0, 10)         | 426.61            | 4215     | 255.95                  | 4254     |
> |          |-------------|-------------------|--------------------|----------|--------------------------|----------|
> |          | CS          | 𝒩(0, 1)          | **69.76**         | 2823     | **59.59**               | 2833     |
> |          |             | 𝒩(0, 4)          | **252.92**        | 4224     | **210.71**              | 2832     |
> |          |             | 𝒰(0, 10)         | **299.80**        | 4215     | **246.57**              | 4234     |
> |

---

> ### Author Response · Authors · 2024-11-21
> **Rebuttal by authors**
>
> > **[Q7] The merge-and-reduce framework is crucial for scaling the coreset method, but the paper provides limited guidance on its implementation and scalability implications: Although the paper briefly mentions time complexity, a more detailed breakdown of the merge-and-reduce framework’s complexity across layers and for different dataset sizes would be helpful.**
>
>
> Thank you for your valuable suggestions. Below is a more detailed analysis of the time complexity for Algorithm 2 .
>
> Each layer of Algorithm 2 performs multiple computations of Algorithm 1 parallelly, where the input data size of Algorithm 1 is $ O(\tau) $. Consequently, the time complexity of per layer is $ \tilde{O}(2^{2 \cdot \text{ddim}} \cdot \tau \cdot T(n)) $.
>
> Given that the input size for Algorithm 1 is $ O(\tau) $ and the output size （coreset size） is $ s $, the compression ratio of per layer is $ \frac{\tau}{s} $. As a result, the tree can have at most $ H = \log_{\frac{\tau}{s}} \dfrac{|Q|}{s}$ layers. Therefore, the overall time complexity of the framework is $ \tilde{O}(2^{2 \cdot \text{ddim}} \cdot \tau \cdot T(n)) $.
>
> ---
>
> ---
>
> > **[Q8] There is limited discussion on how the framework’s parallelization could be optimized or applied to larger datasets. This discussion would be especially useful for practitioners seeking to apply the method in large-scale real-world settings.**
>
>
> Thank you for your valuable suggestions.
>
> Without utilizing the parallel framework, the time complexity of computing a coreset for dataset $Q$ is $\tilde{O}(2^{2\cdot ddim}\cdot |Q|\cdot T(n))$, which linearly depends on the dataset size and poses significant challenges when handling large datasets. By employing our framework, the time complexity $\tilde{O}(2^{2\cdot ddim}\cdot \tau \cdot T(n))$ is effectively reduced, making the computational efficiency independent of dataset size, resulting in substantial improvements.
>
> Moreover, communication complexity is efficient. Specifically, our coreset is a subset of the original dataset, allowing us to transmit only the indexes of the CO instance items rather than the data items themselves. This significantly reduces transmission costs. As a result, the additional transfer complexity introduced by our merge-and-reduce framework in Algorithm 2 is, in practice, minimal and unlikely to pose a substantial overhead.
>
> Our Algorithm 2 is efficient in both time complexity and communication complexity.  This makes the framework particularly efficient for distributed and large-scale applications.
>
> We will include a more detailed discussion in the revised manuscript to clarify these aspects and better address your concerns. Thank you again for your constructive comments.

---

> > ### Author Response · Authors · 2024-11-21
> > **Rebuttal by authors**
> >
> > > **[Q9] While the paper claims improved robustness to distribution shifts through the use of coresets, this aspect is not rigorously analyzed or compared. The paper could benefit from quantifying the robustness improvements by comparing the method’s performance across significantly different distributions and measuring accuracy decay.**
> >
> >
> > Thank you for your valuable suggestions. For convenience, we use Table 1 as an example to show the performance on robustness to distribution shifts.
> >
> > In Table 1, we demonstrate the robustness of our method by evaluating performance on test datasets drawn from distributions significantly different from the training distribution. Specifically, the training data is sampled from a normal distribution $N(0,1)$, while the test data is sampled from normal distributions $N(0,1),N(0,4)$ and a uniform distribution $U(0,10)$.  The distributions $N(0,4), U(0,10)$ are significantly different from the training distribution $N(0,1)$, which represent substantial distribution shifts. The results in Table 1 show that our method consistently outperforms the baselines, demonstrating its robustness to distribution shifts.
> >
> >
> > ---
> >
> > ---
> >
> >
> >
> > **Table 1:** Comparison of Uniform Sampling and Coreset Method on Test Data TSP100-2D
> >
> > ---
> >
> > | Sample Size | Method      | Test Distribution  | Greedy Length (↓) | Greedy Time (↓) | Greedy+2-opt Length (↓) | Greedy+2-opt Time (↓) |
> > |-------------|-------------|--------------------|--------------------|-----------------|-----------------|-------------|
> > | 4003    | US          | 𝒩(0, 1)           | 22.34             | 378             | 18.92                   | 387                   |
> > |             |             | 𝒩(0, 4)           | 101.95            | 379             | 69.28                   | 395                   |
> > |             |             | 𝒰(0, 10)          | 119.78            | 380             | 82.59                   | 395                   |
> > |             |-------------|--------------------|--------------------|-----------------|----------------|-------------|
> > |             | CS          | 𝒩(0, 1)           | 22.21             | 372             | 18.87                   | 379                   |
> > |             |             | 𝒩(0, 4)           | 80.63             | 372             | 67.92                   | 379                   |
> > |             |             | 𝒰(0, 10)          | 94.73             | 373             | 80.64                   | 377                   |
> > |-------------|-------------|--------------------|--------------------|-----------------|-----------------|-------------|
> > | 8245    | US          | 𝒩(0, 1)           | 22.12             | 377             | 18.87                   | 388                   |
> > |             |             | 𝒩(0, 4)           | 83.17             | 377             | 68.13                   | 378                   |
> > |             |             | 𝒰(0, 10)          | 97.31             | 377             | 80.80                   | 387                   |
> > |             |-------------|--------------------|--------------------|-----------------|----------------|-------------|
> > |             | CS          | 𝒩(0, 1)           | 21.79             | 366             | 18.84                   | 383                   |
> > |             |             | 𝒩(0, 4)           | 78.72             | 372             | 67.79                   | 378                   |
> > |             |             | 𝒰(0, 10)          | 92.99             | 374             | 80.35                   | 377                   |
> > |-------------|-------------|--------------------|--------------------|-----------------|----------------|-------------|
> > | 12951  | US          | 𝒩(0, 1)           | 21.99             | 390             | 18.87                   | 377                   |
> > |             |             | 𝒩(0, 4)           | 80.78             | 384             | 67.94                   | 379                   |
> > |             |             | 𝒰(0, 10)          | 95.01             | 369             | 80.60                   | 379                   |
> > |             |-------------|--------------------|--------------------|-----------------|---------------|-------------|
> > |             | CS          | 𝒩(0, 1)           | 21.57             | 372             | 18.81                   | 382                   |
> > |             |             | 𝒩(0, 4)           | 77.80             | 369             | 67.58                   | 379                   |
> > |             |             | 𝒰(0, 10)          | 92.01             | 378             | 80.23                   | 375                   |
> > |
> >
> > Additional results supporting this conclusion can be found in Tables 1, 2, 4, 6, 7, 10 and 11. Across these evaluations, our approach consistently demonstrates superior performance compared to the baselines, even under significant distribution shifts. We hope these findings address your concern and illustrate the robustness improvements achieved by our method.

---

> > > ### Author Response · Authors · 2024-11-24
> > >
> > > Thank you for your dedication and interest in our paper. As the author and reviewer discussion period approaches its end, we are curious to know your thoughts on our rebuttal and whether you have any additional questions.
> > > We hope to have the opportunity to further improve the paper based on your additional suggestions.

---

> > > > ### Comment · Reviewer_wmHE · 2024-11-26
> > > >
> > > > Thank the authors for providing additional experiments. I have raised my score.

---

### Official Review · Reviewer_CqAL · 2024-11-01

**Soundness:** 2
**Presentation:** 3
**Contribution:** 2
**Rating:** 6
**Confidence:** 3

**Summary:**

In this article, the author(s) develop a framework for accelerating neural combinatorial optimization (CO) methods. The basic idea is to construct a small-size coreset to represent the whole data set, and only train models on the coreset. The coreset is constructed based on a clustering algorithm and the Wasserstein distance under rigid transformations (RWD).

**Strengths:**

The overall motivation of this article is clear. Efficiently solving CO problems are of great importance, and this article provides a potential direction.

**Weaknesses:**

1. Although the main topic of this article is about combinatorial optimization (CO) problems, it seems that the whole article ignores the structure and discreteness of CO problems, and only considers their graph embedding. Then the author(s) only consider objects that lie in an Euclidean space, which leaves me the impression that the proposed method is developed for continuous problems, and CO only appears in the preprocessing stage (i.e., converting a CO instance into a continuous object). I am not sure if this is the proper way to deal with CO problems, but at least the author(s) should discuss the relationship between the original CO problem and the embedding. For example, is there any information loss after converting into embeddings? Are the results sensitive to the choice of embedding methods?

2. I think the equation in Definition 2.1 is incorrect. The $C_{ij}$ term should be $P_{ij}$, and the cost matrix $C$ is never used.

3. It seems that the Wasserstein distance under rigid transformation (RWD) is a core component of the proposed method, but I do not see the method to compute it. In Remark 3.1, the author(s) claim that RWD can be solved within $\tilde{O}(n^2)$ time, but I do not see why. Computing RWD should be much more difficult than the classical optimal transport (OT), as it involves optimization over an additional object $e$. But even for OT, I wonder how the complexity $\tilde{O}(n^2)$ can be achieved without using approximation methods such as the entropic regularization, as it is well known that a linear programming solver for OT takes $O(n^3\log(n))$ [1].

4. If computing RWD is expensive, then I wonder whether it is meaningful to construct the coreset at all. The author(s) are suggested comparing the cost of training on the whole data set and the cost of constructing the coreset plus the time of training on the coreset.

[1] Pele, O., & Werman, M. (2009). Fast and robust earth mover's distances. In 2009 IEEE 12th international conference on computer vision.

**Questions:**

See the "Weaknesses" section.

---

> ### Author Response · Authors · 2024-11-20
> **Rebuttal by authors**
>
> > **[Q1] Although the main topic of this article is about combinatorial optimization (CO) problems, it seems that the whole article ignores the structure and discreteness of CO problems, and only considers their graph embedding. Then the author(s) only consider objects that lie in an Euclidean space, which leaves me the impression that the proposed method is developed for continuous problems, and CO only appears in the preprocessing stage (i.e., converting a CO instance into a continuous object). I am not sure if this is the proper way to deal with CO problems, but at least the author(s) should discuss the relationship between the original CO problem and the embedding. For example, is there any information loss after converting into embeddings? Are the results sensitive to the choice of embedding methods?**
>
>
> Thank you for your insightful questions. We apologize for the lack of clarity on the relationship between CO problems and graph embedding in Euclidean space. Details are as follows.
>
>
> Our method first extracts the graph structure induced by the CO instance and represents it by a graph metric space, where each point in this space reflects node-specific information, and edge relationships are captured through the corresponding shortest-path metric. We then apply graph embedding techniques to map this graph metric space into Euclidean space, aiming to preserve inter-point distances closely. In this embedding, each node in the original graph is represented as a discrete point in Euclidean space, and edge information is encoded in Euclidean distances between these points.
>
> In summary, we ultimately represent the graph as a discrete set of points in Euclidean space. The graph structure (nodes and edges) is described by the points and their distances in Euclidean space.
> Through the graph embedding above, we can focus solely on the set of points in Euclidean space.
>
> Thank you again for your comments, which will help us improve the clarity and rigor of our paper.
>
>
>
>
> > **[Q2] I think the equation in Definition 2.1 is incorrect. The term should be , and the cost matrix is never used.**
>
> Thanks for your careful review. We have corrected the equation in Definition 2.1 accordingly and clarified the use of the cost matrix in the revised version.
>
> > **[Q3] It seems that the Wasserstein distance under rigid transformation (RWD) is a core component of the proposed method, but I do not see the method to compute it. In Remark 3.1, the author(s) claim that RWD can be solved within time, but I do not see why. Computing RWD should be much more difficult than the classical optimal transport (OT), as it involves optimization over an additional object. But even for OT, I wonder how the complexity can be achieved without using approximation methods such as the entropic regularization, as it is well known that a linear programming solver for OT takes [1].**
>
>
>
>
> Thank you very much for your suggestion. I apologize for the lack of clarity in the manuscript. I have provide more analysis on time complexity about optimal transport (OT) in Appendix, including the work [2] demonstrating an $\tilde{O}(n^2/\epsilon_+)$ time complexity for OT computation.  We regard the additive error $\epsilon_+$ as a constant, thus the time complexity is $\tilde{O}(n^2)$.
>
>
> Moreover, the algorithm for computing RWD is added to the Appendix. We compute the RWD by alternating between optimizing the coupling matrix and the rigid transformation, which is a heuristic method. We assuming that the point dimension $d$ and the number of iterations are constants. For obtaining the coupling matrix, we solve an OT problem, which, based on the work [2], can indeed be computed in $\tilde{O}(n^2)$ time. The rigid transformation, on the other hand, involves solving an orthogonal Procrustes problem, which has a time complexity of $O(n^2d+nd^2+d^3)$.
> Thus, the overall complexity of this heuristic method remains $\tilde{O}(n^2)$.
>
> I hope this clarifies the approach in my revised version. More details for computing RWD are in Appendix.
>
> **References**
>
> [2] Jambulapati A, Sidford A, Tian K. A direct tilde {O}(1/epsilon) iteration parallel algorithm for optimal transport[J]. Advances in Neural Information Processing Systems, 2019, 32.

---

> ### Author Response · Authors · 2024-11-20
> **Rebuttal by authors**
>
> > **[Q4] If computing RWD is expensive, then I wonder whether it is meaningful to construct the coreset at all. The author(s) are suggested comparing the cost of training on the whole data set and the cost of constructing the coreset plus the time of training on the coreset.**
>
> Thank you for your insightful advice. You are absolutely correct that computing the exact solution for RWD can be computationally intensive. However, we have found that obtaining a high-quality heuristic solution is relatively straightforward and efficient, making it feasible for practical applications.
>
> Moreover, **Table 5,9,13,19** in our Appendix
> present a comparison of the time cost for training on the full dataset versus constructing the coreset and training on it. For convenience, we take Table 5 as an example to show our performance,  which demonstrates the time efficiency of our algorithm.
>
> ---
>
> ---
>
> **Table 5:** Time statistics for different phases of training on TSP100-2D-𝒩(0, 1)
> | Method | Sample size | Labeling time | Coreset Time | Training Time | Total time |
> |--------|-------------|---------------|--------------|---------------|------------|
> | Org    | 128000      | 4709          | -            | 28563         | 33272      |
> |--------|-------------|---------------|--------------|---------------|------------|
> |        | 4003        | 147           | -            | 1894          | 2041       |
> | US     | 8245        | 304           | -            | 2862          | 3166       |
> |        | 12951       | 475           | -            | 4014          | 4489       |
> |--------|-------------|---------------|--------------|---------------|------------|
> |        | 4003        | 145           | 691          | 1731          | 2567       |
> | CS     | 8245        | 305           | 1086         | 2747          | 4138       |
> |        | 12951       | 474           | 1283         | 3751          | 5508       |
>
>
> In addition, our coreset only needs to be computed once, after which it can be used repeatedly to train different models and fine-tune parameters. Even if the coreset computation is time-consuming, it is still valuable as it helps save storage space.
>
> Thank you again for the valuable suggestion!

---

> > ### Author Response · Authors · 2024-11-24
> >
> > Thank you for your dedication and interest in our paper. As the author and reviewer discussion period approaches its end, we are curious to know your thoughts on our rebuttal and whether you have any additional questions.
> > We hope to have the opportunity to further improve the paper based on your additional suggestions.

---

> ### Comment · Reviewer_CqAL · 2024-11-25
>
> First of all, I would like to thank the author(s) for the various clarifications, and some of my concerns have been addressed. Based on this, I decide to raise my score. However, I think there are still some issues not fully resolved by the current manuscript.
>
> For example, the relevance between two CO instances is mostly determined by their Euclidean embeddings, but the quality and effectiveness of this transformation does not have a strong guarantee. The author(s) are advised to discuss potential information loss when converting CO problems into embeddings, and do some sensitivity analysis using different embedding methods. Also, it leaves me the impression that the rest part of this article is only loosely connected with CO.
>
> The second problem is that substantial discussions need to be added for RWD, especially its computation. If my understanding is correct, solving RWD is not a convex problem, so if the author(s) use heuristic methods, then its accuracy also need to be taken into account.

---

> ### Author Response · Authors · 2024-11-30
> **Rebuttal by authors**
>
> > **[Q1] For example, the relevance between two CO instances is mostly determined by their Euclidean embeddings, but the quality and effectiveness of this transformation does not have a strong guarantee. The author(s) are advised to discuss potential information loss when converting CO problems into embeddings, and do some sensitivity analysis using different embedding methods.**
>
> Thank you for your valuable suggestion. In our approach, we primarily use graph embedding techniques to select diverse graph data for training. However, we still train on the data items selected from the original dataset, rather than using the embedded data. Therefore, some level of information loss during the embedding process is acceptable. We do not have strict requirements for embedding accuracy, as long as the preserved information is sufficient to to help select diverse data.
>
> We take Maximum Independent Set (MIS) as an example to demonstrate the effectiveness of our method. Figure 27 presents the experimental results with different graph embedding techniques. While there are some variations among the various graph embedding methods, all of them outperform the baseline method (uniform sampling).
>
> ----
>
> ---
>
> **Table 27:** Comparison of uniform sampling and our coreset method
> with different graph embedding techniques  on test data from different distributions.
> CS-spring is the embedding technique based on force-directed representation;
> CS-spectral is the spectral embedding technique;
> CS-MDS is the embedding technique based on  multidimensional scaling.
>
> ----
>
> | Sample size | Method     | Test distribution  | Size $(\uparrow)$ | Time $(\downarrow)$ |
> |-------------|------------|--------------------|-------------------|---------------------|
> | 4010        | US         | ER-[400-500]       | 27.40             | 133                 |
> |             |            | ER-[700-800]       | 30.36             | 392                 |
> |             |            | ER-[1400-1500]     | 34.05             | 1361                |
> |-------------|------------|--------------------|-------------------|---------------------|
> | 3973        | CS-spring  | ER-[400-500]       | 28.46             | 135                 |
> |             |            | ER-[700-800]       | 30.89             | 389                 |
> |             |            | ER-[1400-1500]     | 34.25             | 1361                |
> |-------------|------------|--------------------|-------------------|---------------------|
> | 3994        | CS-spectral| ER-[400-500]       | 27.68             | 132                 |
> |             |            | ER-[700-800]       | 30.43             | 391                 |
> |             |            | ER-[1400-1500]     | 34.14             | 1362                |
> |-------------|------------|--------------------|-------------------|---------------------|
> | 4010       | CS-MDS     | ER-[400-500]       | 28.43             | 132                 |
> |             |            | ER-[700-800]       | 31.10             | 389                 |
> |             |            | ER-[1400-1500]     | 34.52             | 1361                |

---

> ### Author Response · Authors · 2024-11-30
> **Rebuttal by authors**
>
> > **[Q2] Also, it leaves me the impression that the rest part of this article is only loosely connected with CO.**
>
>
> Thank you very much for your valuable suggestion. You are absolutely right, and your observation highlights another key strength of our approach.
>
> Our framework is not limited to combinatorial optimization (CO) problems; it is equally effective for other graph-structured datasets and related classification tasks. The reason we used CO as an example is that we are more familiar with this area, which allowed us to better demonstrate the application of the framework.
> However, the framework is by no means restricted to CO; it can be applied to fields like chemical analysis and biology, where it aids in efficient data pruning and labeling. This versatility is precisely why we have referred to it as a "framework." Actually, CO is just one of many problems that the framework can address, and its applications extend far beyond CO, encompassing a wide range of domains.
>
> ----
>
> ----
>
>
> > **[Q3]The second problem is that substantial discussions need to be added for RWD, especially its computation. If my understanding is correct, solving RWD is not a convex problem, so if the author(s) use heuristic methods, then its accuracy also need to be taken into account.**
>
>
> Thank you very much for your valuable suggestion. Regarding the accuracy of RWD calculation, it is indeed a complex non-convex problem.
>
> However, in practical applications, we typically do not require an exact solution, but rather seek an approximate one, which is sufficient for our purposes.
> Therefore, we provide a heuristic algorithm for solving RWD (outlined in Algorithm 3 of Appendix B), along with a detailed analysis of its performance.
> Our RWD uses an iterative optimization approach. For general optimization algorithms, the changes in the initial iterations are usually significant, and later iterations result in smaller adjustments in the local region. Therefore, after a few rounds of iteration, our RWD algorithm can obtain a relatively good solution.
>
> In our coreset method, RWD is primarily used to help select diverse data items, so some loss of accuracy is therefore acceptable. As long as it can roughly describe the differences between the data, small deviations in accuracy do not significantly affect the final results. This is similar to many clustering problems, which, although NP-hard in theory, can often be solved efficiently in practice and are widely applied.
>
> ----
>
>
>
> Moreover, our framework is highly flexible, and RWD is just one example chosen to illustrate the graph dataset compression and pruning problem. We can easily replace RWD with other suitable distance metrics, such as WD (Wasserstein Distance) or GWD (Gromov-Wasserstein Distance), depending on the specific requirements.
>
> If a stricter theoretical result is needed, we can replace RWD with WD. RWD helps to mitigate the impact of rigid transformations, whereas WD is more theoretically rigorous. If the impact of rigid transformations is not a concern, WD would be a good alternative.
>
> If capturing the full structural information of the graph is essential and one is willing to accept higher time complexity, we can also use GWD. In this case, we do not need to embed the graph into Euclidean space but can directly compute the distance between the corresponding  graph metrics of CO instances. However, the time complexity of GWD is much higher than that of RWD, with time complexity of $O(n^3)$[1].
>
> We chose to use RWD primarily to balance efficiency with the consideration of rigid transformation effects on the graph dataset.
>
> In future work, we will further explore the effects of WD and GWD. Once again, we appreciate your attention and insightful suggestions.

---

> > ### Author Response · Authors · 2024-12-02
> >
> > Thank you for your insightful questions. We have provided further responses and look forward to your guidance.

---

> > > ### Author Response · Authors · 2024-12-03
> > >
> > > Dear reviewer,
> > >
> > > Thank you for your insightful guidance on my paper. As the rebuttal phase deadline is approaching in a few hours, I am eager to receive any further suggestions and feedback you may have to strengthen our submission.
> > >
> > > Thank you again for your time and support.
> > >
> > > Best regards, The Authors

---

> > > > ### Comment · Reviewer_CqAL · 2024-12-03
> > > >
> > > > Thanks for the additional explanations. I think most of my questions have been answered, and I would like to further raise my score.

---

### Official Review · Reviewer_SBLn · 2024-11-04

**Soundness:** 3
**Presentation:** 2
**Contribution:** 3
**Rating:** 6
**Confidence:** 3

**Summary:**

The paper presents a novel approach to enhance neural combinatorial optimization (NCO) by introducing Wasserstein-based coresets, which efficiently compress large datasets into smaller, representative proxies. By modeling combinatorial optimization (CO) instances as probability measures and utilizing a specialized Wasserstein distance under rigid transformations (RWD), the authors quantify differences between CO instances effectively. To address the computational challenges of constructing coresets for large datasets, they adapt the merge-and-reduce framework, enabling parallel processing and theoretical guarantees of representation quality. Additionally, the proposed training framework leverages these coresets to reduce computational and storage requirements while maintaining and even improving robustness to distribution shifts between training and testing data. Experimental results on Traveling Salesperson Problem (TSP) and Maximum Independent Set (MIS) instances demonstrate that their method outperforms uniform sampling and other existing techniques, achieving better performance and enhanced robustness with reduced resource usage.

**Strengths:**

The paper effectively addresses key challenges in neural combinatorial optimization (NCO) by introducing Wasserstein-based coresets that reduce dataset size without compromising essential information. The use of Wasserstein distance under rigid transformations (RWD) provides a robust metric for comparing combinatorial optimization instances, enhancing the method's ability to handle distribution shifts. By adapting the merge-and-reduce framework, the authors achieve scalable and parallelizable coreset construction, making the approach feasible for large datasets. Theoretical guarantees ensure that the coresets accurately represent the original data, while the proposed training framework demonstrates reductions in computational and storage requirements. Experimental results on the Traveling Salesperson Problem (TSP) and Maximum Independent Set (MIS) depict the method's superior performance and increased robustness compared to uniform sampling.

**Weaknesses:**

While the paper presents a rigorous analysis of Wasserstein-based coresets to enhance NCO, it exhibits several weaknesses. In particular, the English language is somewhat poor, leading to unclear statements $-$ for instance, "Therefore, how to train a competitive model by using limited resources while guaranteeing its robustness to distribution shift is a deserving problem" is awkwardly phrased. Technically, the justification for modeling combinatorial optimization instances as probability measures using Rigid Wasserstein Distance (RWD) seems incomplete, especially since the assumptions like low doubling dimension may not hold in practical, high-dimensional settings (e.g., problems with large graphs or high-dimensional feature spaces). Despite the reduced algorithmic complexity, it is unclear whether the proposed merge-and-reduce framework (to accelerate the coreset construction algorithm) may introduce additional practical overhead due to the increased cost of data transfer (via partitioning and merging) and possible synchronization delays.

**Questions:**

Does the proposed merge-and-reduce framework to accelerate coreset construction introduce additional practical overhead due to increased data transfer costs (via partitioning and merging) and possible synchronization delays?

---

> ### Author Response · Authors · 2024-11-20
> **Rebuttal by authors**
>
> > **[Q1] Does the proposed merge-and-reduce framework to accelerate coreset construction introduce additional practical overhead due to increased data transfer costs (via partitioning and merging) and possible synchronization delays?**
>
>
> Thank you very much for your insightful question. Your question helped us recognize an important advantage of our Algorithm 2 regarding communication efficiency.
>
> Specifically, our coreset is a subset of the original dataset, allowing us to **transmit only the indexes of the CO instance items** rather than the data items themselves. This significantly reduces transmission costs.  As a result, the additional transfer complexity introduced by our merge-and-reduce framework in Algorithm 2 is, in practice, minimal and unlikely to pose a substantial overhead.
>
>
> > **[Q2] Technically, the justification for modeling combinatorial optimization instances as probability measures using Rigid Wasserstein Distance (RWD) seems incomplete, especially since the assumptions like low doubling dimension may not hold in practical, high-dimensional settings (e.g., problems with large graphs or high-dimensional feature spaces).**
>
>
> Thank you very much for your insightful question.
>
> We acknowledge that the low doubling dimension assumption is primarily intended to facilitate theoretical analysis. Analyzing the general case without assuming a low doubling dimension would be significantly more challenging.
>
> However, in practical applications, it is often not necessary to know the exact value of the doubling dimension in advance. Typically, we begin by experimenting with relatively small values, as demonstrated in our study where we set the low doubling dimension as $ddim=1$. In practice, even if we cannot rigorously prove that the data satisfies low doubling dimension assumption, this generally does not impact the effectiveness of our experimental results.
>
>
>
>
>
> > **[Q3] In particular, the English language is somewhat poor, leading to unclear statements--for instance, "Therefore, how to train a competitive model by using limited resources while guaranteeing its robustness to distribution shift is a deserving problem" is awkwardly phrased.**
>
> Thank you for your careful review! I will continue refining the language to make the expressions clearer and more natural.

---

> > ### Author Response · Authors · 2024-11-24
> >
> > Thank you for your dedication and interest in our paper. As the author and reviewer discussion period approaches its end, we are curious to know your thoughts on our rebuttal and whether you have any additional questions.
> > We hope to have the opportunity to further improve the paper based on your additional suggestions.

---

> > ### Comment · Reviewer_SBLn · 2024-11-25
> >
> > Thank you for the provided clarifications. I have decided to raise my score, although I still have major concerns about the quality of language in the paper. As a brief aside, I would advise the authors to remove the bolded terms (e.g., "first", "Then", "next") in the abstract. This format is somewhat nonstandard and interrupts the flow of the paper.

---

> > > ### Author Response · Authors · 2024-12-04
> > >
> > > Thank you for your suggestions. I have made revisions in the latest version that I submitted.

---

### Meta-Review · Area_Chair_9Jom · 2024-12-18

**Metareview:**

The paper presents an approach for efficient neural combinatorial optimization (NCO) by leveraging Wasserstein-based coresets. Key components of the method include:

- Representing NCO Instances as probability measures;
- Using a Wasserstein distance under rigid transformations (RWD) to quantify similarity;
- A scalable merge-and-reduce framework for parallelized coreset creation.

Empirical results demonstrate superior generalization and computational efficiency compared to baseline methods.

Reviewers generally agree on the paper’s originality, clarity, and significance, and uniformly recommended acceptance. While there were clarification questions and requests for additional experimental results, most of the concerns have been addressed during the discussion period. Overall, the paper introduces a nice and novel method to address an important practical problem with sufficient experimental verification. Hence I would recommend acceptance.

For the revised version, I’d encourage the authors to improve further based on the reviewer feedback, including, 1) adding clarification when needed, 2) better addressing the computation cost of RWD,  3) incorporating additional experiments as suggested by reviewer wmHE and others.

**Additional Comments On Reviewer Discussion:**

During the discussion, the authors provided clarification to reviewers' questions and added more experimental results to address concerns (mainly from reviewer wmHE) on various aspects of experimental verification. All reviewers are satisfied with the responses.

---

### Decision · Program_Chairs · 2025-01-22

Accept (Poster)